# Real-space heterogeneous reconstruction, refinement, and disentanglement of CryoEM conformational states with HetSIREN

David Herreros [1] ✉, Carlos Perez Mata [1,2], Chari Noddings[3], Deli Irene[4], James Krieger [1], David A. Agard[5,6], Ming-Daw Tsai [4], Carlos Oscar Sanchez Sorzano [1,7] & Jose Maria Carazo [1,7]

Single-particle analysis by Cryo-electron microscopy (CryoEM) provides direct access to the conformations of macromolecules. Traditional methods assume discrete conformations, while newer algorithms estimate conformational landscapes representing the different structural states a biomolecule explores. This work presents HetSIREN, a deep learning-based method that can fully reconstruct or refine a CryoEM volume in real space based on the structural information summarized in a conformational latent space. HetSIREN is defined as an accurate space-based method that allows spatially focused analysis and the introduction of sinusoidal hypernetworks with proven high analytics capacities. Continuing with innovations, HetSIREN can also refine the images' pose while conditioning the network with additional constraints to yield cleaner high-quality volumes, as well as addressing one of the most confusing issues in heterogeneity analysis, as it is the fact that structural heterogeneity estimations are entangled with pose estimation (and to a lesser extent with CTF estimation) thanks to its decoupling architecture.

Cryo-electron microscopy (CryoEM) Single Particle Analysis (SPA)[1] ability to capture individual images of biological samples brings to light the challenging capacity to identify several conformational and/or compositional states from the acquired image dataset. Classically, compositional heterogeneity and conformational heterogeneity/flexibility have been addressed through rounds of 3D classification[2] under the assumption that macromolecules adopt a discrete set of states. Discrete classification has been applied successfully and is at the heart of the so-called "Resolution Revolution"[3]. However, the discrete approach introduces a series of limitations that arise from the assumptions on which it is based. Removing this discretization constraint is methodologically a very challenging task. However, the pay-offs are clear in obtaining richer conformational landscapes than is currently done, providing improved algorithmic stability and objectivity, removing assumptions not supported by the data, and streamlining the analysis process without trial error tests and decisions on the quality and number of classes.

Algorithms for identifying continuous heterogeneity from particle images were first introduced in 2014[4,5]. More recently, CryoDRGN[6] introduced the concept of heterogeneous reconstruction, applying advanced neural network techniques to address the approximation of the conformational continuum through the decodification of per-particle structural states. Similarly, other approaches have been proposed to tackle the heterogeneous reconstruction problem, such as heterogeneous reconstruction with Gaussian Mixtures[7] or heterogeneous reconstruction with a priori information on conformational latent space[8], among others.

[1]Centro Nacional de Biotecnologia-CSIC, C/ Darwin, 3, Cantoblanco Madrid, Spain. [2]PKF Attest innCome, Orense 81, Madrid, Spain. [3]Altos Labs, Redwood City, CA, USA. [4]Institute of Biological Chemistry, Academia Sinica, Taipei, Taiwan. [5]Department of Biochemistry & Biophysics, University of California, San Francisco, CA, USA. [6]Chan Zuckerberg Imaging Institute, Redwood City, CA, USA. [7]These authors jointly supervised this work: C.O.S. Sorzano and J.M. Carazo. ✉e-mail: dherreros@cnb.csic.es

In addition to the reconstruction of heterogeneous states, other methods have focused on estimating molecular motions/flexibility using deformation fields. Some methods rely on a neural network to decode the deformation field directly from a latent space representation[9,10], while other approaches have proposed to expand the field on a different basis and then use a reduced set of parameters to estimate the complete field[11,12].

This work proposes a different approximation to the heterogeneous reconstruction problem, moving the reconstruction process ultimately to real space. This approach makes it possible to fine-tune the neural network architecture to improve the quality of decoded volumes, reduce noise overfitting, and perform focused/exclusion heterogeneous reconstructions. We achieve these goals by combining three critical innovations. The first one is introducing (in CryoEM) SIREN activation functions in the network architecture. Indeed, compared to other popular approaches to approximate functions with a decoder architecture, such as ReLU with positional encoding, SIREN activations have been shown to preserve much better the quality and high-frequency features of the original signals fed to the network[13]. The second factor is the effective decoupling of pose and CTF effects from the estimation of conformational landscapes, a key issue considering how intertwined these processes are, as noted in[14]. This second goal is achieved by introducing constraints in latent space relating multiple projections of the same structure from different directions. Finally, we introduce a set of regularizers, including $L_1$ and Total Variation minimization, that helps to obtain high-resolution maps from individual coordinates of conformational space.

The practical result of HetSIREN's capabilities is that conformational landscapes are now substantially better at presenting structurally relevant heterogeneity information, and their exploration can be done at high resolution. This fact can significantly affect many biological systems when rounds of 3D classification lead to reduced data sets and low resolution.

Our major contributions are:

- We propose a encoder decoupling architecture to disentangle the pose and CTF estimation explicitly from the structural information in the structural landscapes, directly tackling one of the primary sources of error heterogeneity algorithms face. This approach generates more understandable, accurate, and interpretable landscapes than standard network architectures.
- Application of meta-sinusoidal layers and hypernetworks to decode high-resolution 3D conformational states with enhanced local resolution and structural features compared to standard reconstruction methods.
- Efficient reconstruction of complete 3D volumes in real space, including the possibility to add structural priors to improve the representation of the electron density maps. To that end, real space constraints are added to explicitly mitigate the noise and negative values in the decoded volumes. In addition, constraints in real space to enhance the continuity and sharpness of the protein signal against the noise are included in the network.
- Possibility to include reconstruction masks to focus on or exclude unwanted structures during the heterogeneous reconstruction process.
- We propose a robust multiresolution training scheme to simplify training on high-resolution data where noise becomes a solid limiting factor.
- We apply these methods to identify multiple conformations of the SARS-CoV-2 Spike protein from single datasets and demonstrate their variation with temperature.

## Results

This section analyzes a simulated dataset, followed by a classical public data set commonly used when presenting heterogeneous reconstruction methods, ending with the presentation of collaborative work on challenging specimens.

All the datasets were analyzed with Scipion 3.8.0 software package. Inside Scipion, CryoSPARC 4.5.1, Relion 4.0, and Xmipp 3.24.12.0 packages were also used to process the data.

### Simulated adenylate kinase landscape and landscape disentanglement

To accurately evade the effect of decoupling pose and CTF from the estimated HetSIREN landscape, we propose a simple and conceptual experiment based on the simulation of an open-to-closed trajectory of the adenylate kinase protein (PDB entry 4AKE) using Normal Mode Analysis with HEMNMA[15]. The simulated trajectory is recovered from the excitation of two modes, leading to a ground truth landscape with a straight-line shape. The trajectory was then sampled to generate a set of 500 projections with uniformly distributed poses and variable CTF information.

The simulated projections were imported into Scipion[16] to train two different HetSIREN networks: a network without pose and CTF decoupling and a network with a pose and CTF decoupling encoder. The resulting landscapes are provided in Fig. 1a, b.

As seen in Fig. 1a, the standard autoencoder architecture does not recover the ground truth landscape. However, it effectively captures the simulated motion along the first principal component of the landscape. In general, this is the type of effect expected to arise on standard heterogeneity algorithms due to unwanted factors that compromise the quality of the latent spaces and significantly limit the interpretability of the landscape.

Figure 1b shows the landscape obtained from the decoupling architecture of HetSIREN. In this case, the new landscape successfully captures a structure resembling the ground truth landscape, mainly arising from a more prominent structural component. Therefore, the combination of the decoupling encoder and the decoder in HetSIREN dramatically aids in the interpretation and understanding of the molecular transition captured in a given dataset.

### Conformational landscape of EMPIAR 10028 dataset

To allow the direct evaluation of HetSIREN compared to other popular heterogeneity tools, we have performed a heterogeneity analysis of EMPIAR-10028[17], which has become one of the de facto standard datasets in the field to address the performance of heterogeneity methods.

EMPIAR-10028 entry corresponds to a CryoEM acquisition of the *P. falciparum* 80S ribosome bound to emetine. The raw data from the database was further processed with Scipion[16], resulting in about 50,000 particles. The workflow within Scipion included several consensus and cleaning steps, trying to reduce unwanted images to a minimal representation and increasing the stability of the angular assignment, shifts, and Contrast Transfer Function (CTF) estimations, which are inputs to the algorithm. It should be noted that most heterogeneity algorithms (but not HetSIREN) treat these inputs as fixed variables, increasing the need to work only with well-curated datasets to avoid misleading conclusions during the heterogeneity analysis.

The particles resulting from the previous analysis were fed to the HetSIREN network during the training phase, followed by the analysis of the latent space encoded from the experimental images. To this end, the latent space mentioned here was explored with the help of interactive tools integrated within the Scipion Flexibility Hub[18]. A landscape visualization and exploration example is provided in Fig. 2. Two landscapes are presented in Fig. 2 in the form of 3D UMAPs[19] obtained from the original 10-dimensional space encoded by the network; the one on the left is without pose and CTF decoupling, while the one on the right implements decoupling. Although a ground truth does not exist for this data set, the landscape after pose and CTF decoupling can be segmented much more easily, a fact that we interpret as an

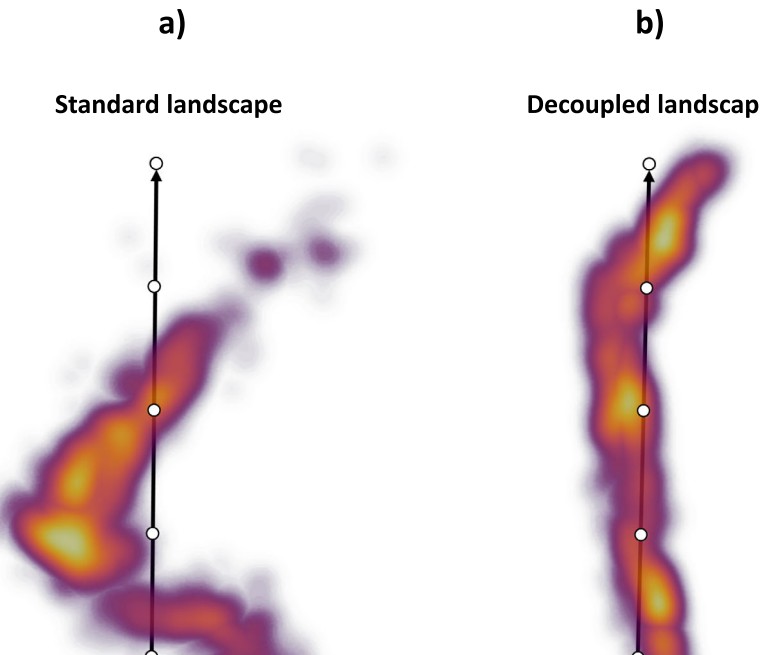

**a) Standard landscape**

**b) Decoupled landscape**

**Fig. 1 | Comparison of HetSIREN standard and pose-CTF decoupled landscapes for the adenylate kinase protein's open-to-close simulated transition.** Ideally, the landscape should approximate the ground-truth trajectory defined as a straight line arising from the excitation of two protein modes. (**a**) shows the landscape obtained with a standard architecture, which suffers from a strong deviation from the ground truth due to the pose and CTF coupling. (**b**) shows the pose and CTF decoupled landscape obtained with HetSIREN. Decoupling the pose and CTF information makes the structural information more prominent, allowing the latent space to approximate the ground-truth conformational landscape well, which should be just a straight line.

enhancement of structural information over positional and CTF "noise". Furthermore, the decoupled landscape was then explored (Fig. 2b) by visualizing a set of maps corresponding to the cluster representatives (centroids) obtained from the 10-dimensional space using the K-Means algorithm, which was later decoded with the network to recover the electron density maps at those points. An initial landscape inspection through the maps shows a non-negligible degree of compositional heterogeneity affecting the ribosome, mainly focused on the 40S ribosomal subunit. Furthermore, it was possible to identify a low-populated state (as shown on Map 12) characterized by a complete lack of the 40S subunit, which is usually not detected due to its low representation (close to 600 particles - around 1.3% of the data). As shown in Fig. 2b, Map 12, HetSIREN successfully identified this evasive state and decoded a map with a resolution similar to those obtained from other more populated landscape regions.

In addition to the compositional heterogeneity analysis of the sample, HetSIREN also allows the identification of the continuous changes captured by the images and the interplay of continuous and compositional changes resulting from their combined influence on the structural characteristics of the complex. In Fig. 3a, b, we provide an example of four decoded maps showing continuous conformational states with and without an extra compositional component, respectively; in each case, the change of conformation is shown by superimposing two maps in two different colors (blue and yellow). The structural change presented in Fig. 3a corresponds to a rotation from left to right of the 40S subunit, one of the main structural changes captured in this dataset. In contrast, the proposed structures in Fig. 3b represent a compositional variation of one of the RNAs found in the ribosomal structure and a significant motion of another RNA generally undetected due to its low resolvability. These examples demonstrate the capacity of HetSIREN to analyze various types of heterogeneity in the same landscape and to

understand the interplay of the different structural modifications that a biomolecule may undergo.

One of the characteristics included in HetSIREN is the possibility of focusing the conformational landscape on a region of interest rather than considering the whole complex during the training phase. This functionality allows us to identify the relevant motions of those regions more easily, which is especially important for small areas since they may have weak relevance in the overall conformational landscape or exclude certain regions from the variability analysis, such as membranes or nanodiscs. In the case currently analyzed, we decided to perform a focused heterogeneity analysis of the ribosomal L1 stalk region, a substantially small area compared to the ribosome but that exhibits a high degree of flexibility. Due to size differences, the contribution of L1 to the general landscape is not predominant, limiting the interpretability of the motions that the L1 stalk undergoes.

To focus on the L1 stalk, we trained a new HetSIREN network with the same particles presented before but providing a spherical mask that covers only the L1 stalk region. HetSIREN can only modify the L1 stalk region thanks to the previous mask, effectively focusing the landscape on this region and excluding the contributions of everything outside the mask. The L1 stalk landscape approximated by HetSIREN and reduced with UMAP is presented in Supplementary fig. 1a. The landscape shows two main motion directions, which can be isolated with PCA as shown in Supplementary fig. 1b. The two main motions correspond to a non-negligible lateral and vertical translation of the L1 stalk, which is more easily identified thanks to the focusing capabilities of HetSIREN.

In addition, in Fig. 4a, b, we provide a comparison of the local resolution computed with DeepRes[20] between a map decoded by HetSIREN and the primary reconstruction obtained from the initial image processing workflow performed in Scipion (that is, the map obtained from the 3D refinement carried out with the complete

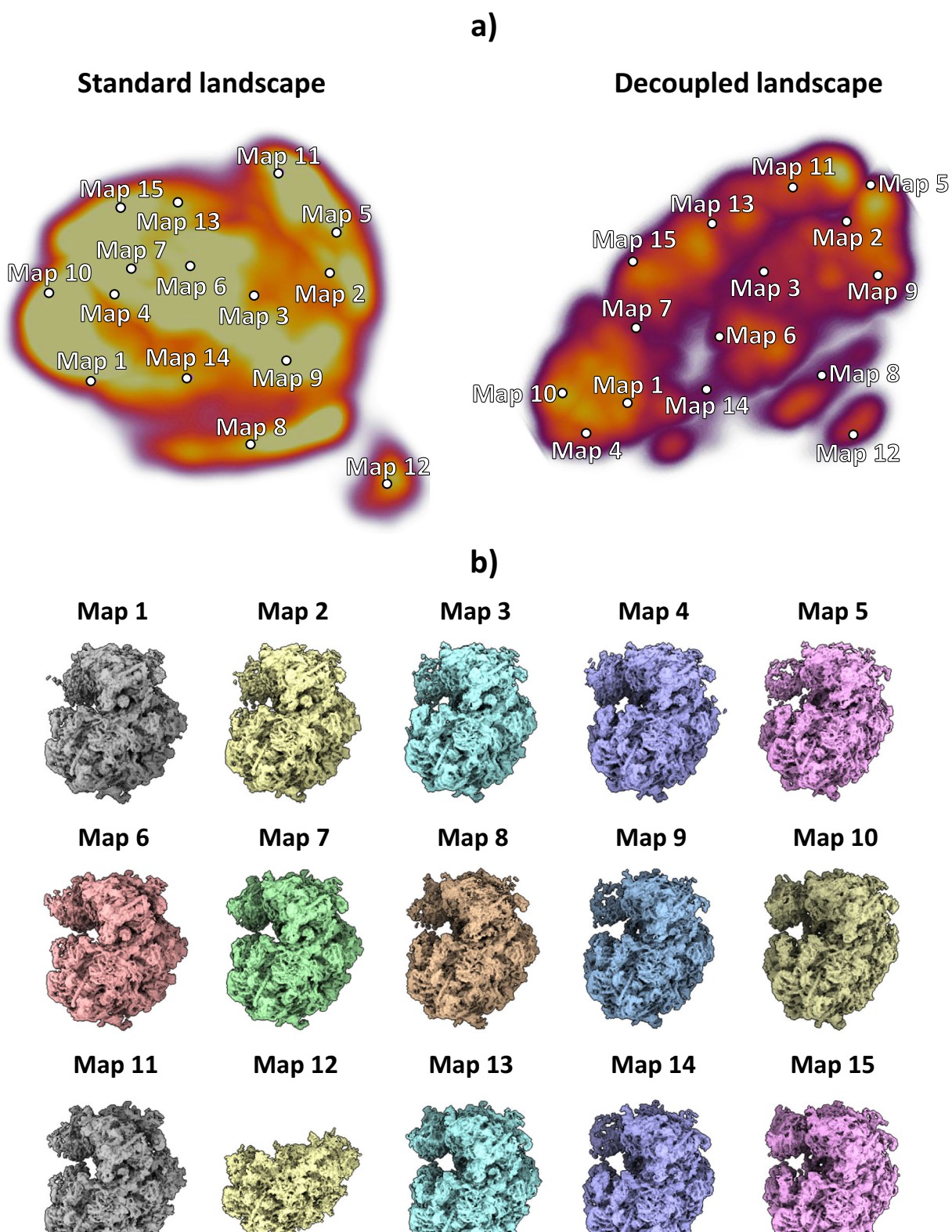

**Fig. 2 | HetSIREN landscape and exploration for Empiar 10028 dataset. (a)** shows the UMAP[19] representation of the landscape obtained from the latent space encoded by HetSIREN from the particle images after training. The landscape without pose and CTF decoupling is presented on the left, while decoupling is implemented on the right landscape. Each dot in the landscape corresponds to the centroid of the cluster representative obtained from a KMeans clustering of the decoupled HetSIREN latent space. **(b)** shows the decoded HetSIREN maps obtained from the decoupled latent space coefficients assigned to every representative, as shown in **(a)** (right). The maps provide a sensible exploration of the different conformational states identified by HetSIREN and a comparison of the structural features learned by the network.

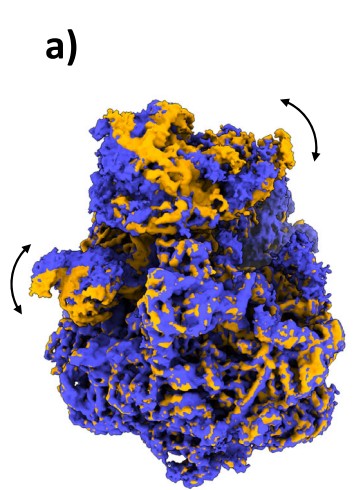
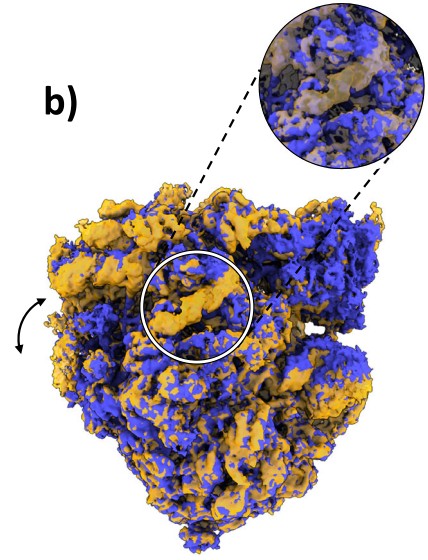

**Fig. 3 | Example of some conformational changes captured in the decoupled HetSIREN landscape presented in Fig. 2a. (a)** shows the main continuous conformational change captured in the dataset, corresponding to a coordinated rotation of the 40S subunit and the R1 stalk of the ribosome. **(b)** shows a compositional variation in one of the ribosomal RNAs (better shown in the magnified image), as well as a large continuous motion of the RNA to the left of the panel, usually not detected due to its low resolvability.

dataset of 50k particles posteriorly used to train HetSIREN). The comparison shows that the HetSIREN decoder does not sacrifice resolution in the decoding process, significantly increasing the local resolution of the map.

## Conformational landscape of the GR:Hsp90:FKBP51 complex

The GR:Hsp90:FKBP51 complex represents a critical molecular assembly regulating the glucocorticoid receptor (GR), a key player in numerous physiological processes, including stress response, metabolism, and immune function. This complex involves the chaperone protein Hsp90 and the immunophilin FKBP51, which together influence the conformation and activity of GR[21]. Unlike its counterpart, FKBP52, which enhances GR activity, FKBP51 acts antagonistically, inhibiting GR's ability to bind ligands and translocate to the nucleus. This inhibition is crucial for maintaining the receptor's homeostasis and responsiveness to hormonal signals. The GR:Hsp90:FKBP51 complex's ability to modulate GR activity has significant implications, as GR dysregulation can lead to various health issues, including immune dysfunction and increased susceptibility to stress-related disorders. Understanding this complex provides insight into potential therapeutic targets for diseases influenced by glucocorticoid signaling.

Given the importance of the GR:Hsp90:FKBP51 complex, we analyzed the intrinsic conformational variability of the dataset presented in ref. 21 with HetSIREN. The dataset included 106884 particles with CTF and angular information already estimated as required by the method.

The images were used to train the network to generate an 8D conformational latent space, posteriorly reduced by PCA[22] to a 3D space for representation purposes. The resulting PCA landscape is provided in Fig. 5a. To gain more insight into the motions detected by HetSIREN, we sampled the leading principal component to generate a set of five different conformational states. The corresponding latent space coordinates were transformed into density maps using the HetSIREN decoder, as presented in Fig. 5b. The figure highlights one of the extreme conformations along the sampled PC 1 axis in a black contour to simplify the understanding of the conformational change (black contour corresponding to Map 5). The results show a significant movement of the GR and FKBP51 regions with respect to the HsP90 protein, resulting in a rotational translation of these two components.

In addition to estimated motions, the quality of decoded HetSIREN volumes was further analyzed and compared with the deposited map from[21] (EMD-29069 [https://www.ebi.ac.uk/emdb/EMD-29069]). Figure 6a shows a direct comparison of the published (left) and HetSIREN (right) maps colored according to their local resolution estimated with DeepRes[20]. The local resolution analysis shows a slight improvement in the resolution of HetSIREN, mainly present in the GR:FKBP51 region. The previous results highlight the ability of HetSIREN to learn and decode high-quality maps, translating into an improved interpretation of the map even in highly dynamic areas.

## Temperature dependence on the conformational landscape of the SARS-CoV-2 Spike protein

The human respiratory coronavirus SARS-CoV-2 is responsible for causing COVID-19, an acute and often severe respiratory illness characterized by intense inflammatory responses and lung damage[23]. Although the virion contains several structural and non-structural proteins, much attention has been directed towards the S (Spike) protein. This glycoprotein forms a trimeric Spike that interacts with the host cell receptor angiotensin-converting enzyme 2 (ACE2) through a mechanism involving the receptor binding domain (RBD) in an equilibrium between RBD opening and closing[24]. Throughout the pandemic, changes in the conformational equilibrium of the RBD have been directly related to the evolution of the virus and the emergence of new variants[25]. These mutations influence the ability of the virus to bind to ACE2 and enter host cells, affecting transmission dynamics and disease severity.

Previous studies have elucidated the impact of the temperature of storage or incubation on the overall integrity and denaturation of the Spike protein[26,27]. It has been reported that the temperature of the protein, equilibrated before and during vitrification, can have a pronounced effect on protein conformation[28], an observation that we wanted to precisely quantify through continuous heterogeneity analysis. In this study, we worked with the Spike protein's beta variant (B.1.351), initially identified in South Africa in the summer of 2020. The cryo-EM datasets for the same sample vitrified at 4 °C and 37 °C according to[29] were acquired separately. Following the acquisition of these datasets and the initial steps of image processing in Scipion, we employed a continuous flexibility analysis using HetSIREN to further

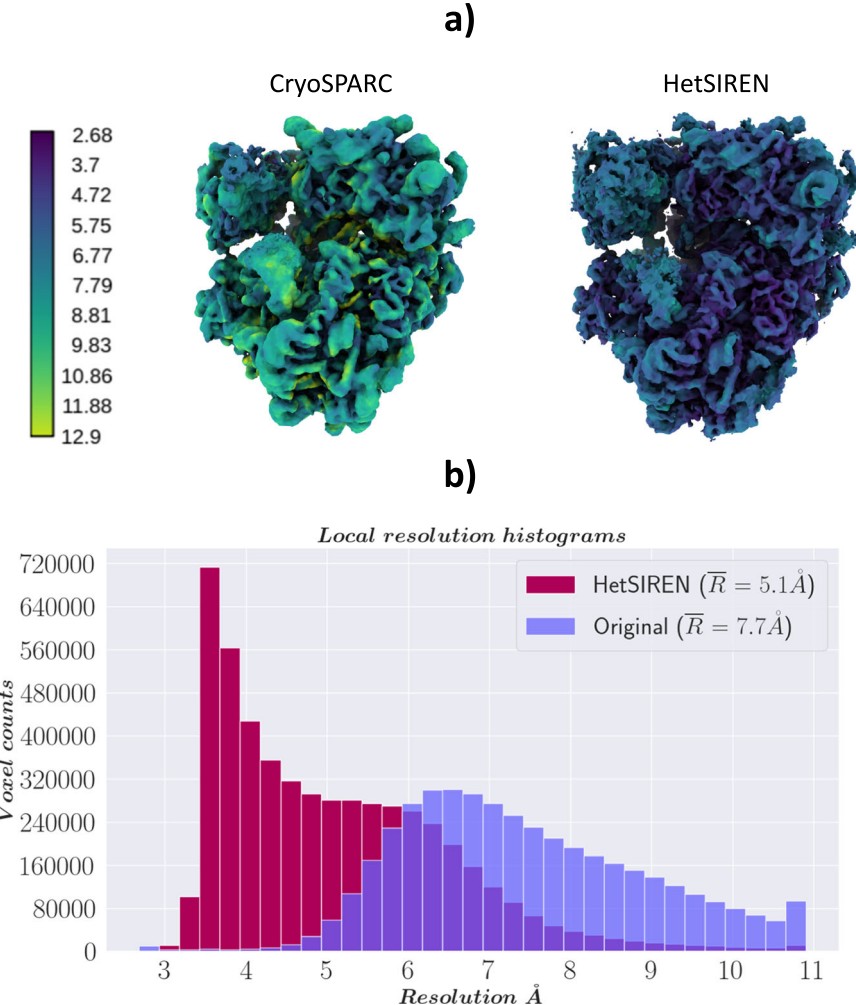

**Fig. 4 | Resolution analysis of HetSIREN map compared against the map refined from the EMPIAR 10028 dataset with CryoSparc[38].** (**a**) shows first the CryoSparc refinement obtained from the 50k particle dataset processed inside Scipion (left) followed by the HetSIREN decoded map (right), both colored according to their local resolution estimated with DeepRes[20]. The comparison shows a significant improvement in the local resolution of the map decoded by HetSIREN. **b** Quantitatively compares the estimated local resolutions based on local resolutions histograms. Similarly to (**a**), the local resolution of HetSIREN shows a strong displacement of the voxel resolutions to the high-resolution domain, translating into an improvement of 2.7 Å in the mean resolution. Source data are provided as a Source Data file.

address the extreme conformational heterogeneity inherent in this sample and characterize subtle conformational changes between the two temperatures.

As previously described, images were used to train the HetSIREN's network to generate an 8D conformational latent space, which was subsequently reduced to a 3D space for representation and analysis purposes using UMAP[19]. To explore the entire conformational landscape and detect all potential conformations, we obtained 20 decoded HetSIREN volumes from a K-Means clustering of the original HetSIREN latent space for each sample.

At 4 °C, our analysis revealed the presence of 1 Up and 2 Up conformations (Fig. 7 and see below), which aligns with previous observations using conventional discrete classification protocols[27]. However, when the sample was at 37 °C, we observed a distinct conformational landscape, predominantly characterized by the 3 Down conformation, which was the only conformation identified by discrete methods (Fig. 8). Nevertheless, our studies benefited from an improved quantification capacity, thanks to the use of advanced analysis tools provided by HetSIREN. This allowed for the additional detection of a reduced contribution from the 1 Up state (Figs. 8, 9b). In addition to the noticeable differences observed in the RBD, we also identified additional dynamic patterns in the N-terminal domain

(NTD), particularly pronounced in the sample at 37°C. As further evidence of the influence of temperature on the Spike protein, we observed that the 1 Up conformation at 37 °C exhibited a less open range of the opening configuration compared to its counterpart at 4 °C (Fig. 9). To quantitatively assess differences in the opening range of the RBD, we generated 29 atomic models that included the 3 Down and 1 Up conformations of the Spike protein at both 37 °C and 4 °C (Supplementary fig. 7). Using the ProDy software tools implemented in Scipion[30], we created an atomic structure ensemble that was subjected to PCA[22]. The RBD opening motions were accurately described by the first principal component (PC1) (Supplementary fig. 7a). For this analysis, we examined several loops in the RBD that exhibit high mobility, as indicated by the PCA results (Supplementary fig. 7b). To quantify the opening range of the RBD, we used centroids of two fixed regions within the core of the Spike protein (S2 domain) as reference points, specifically residues Val991 and Pro1140 from all three chains. We measured the angle between these constant regions and the RBD (Supplementary fig. 7c). The loop spanning residues Thr500-Gly502 provided the most precise description of the differences, showing a clear transition from the 3 Down (6.6° ± 0.3°) to the 1 Up conformation at 37 °C (21.6° ± 0.9°), followed by the 1 Up conformation at 4 °C (24.9° ± 0.7°) (Supplementary fig. 7c inset and Supplementary fig. 7d).

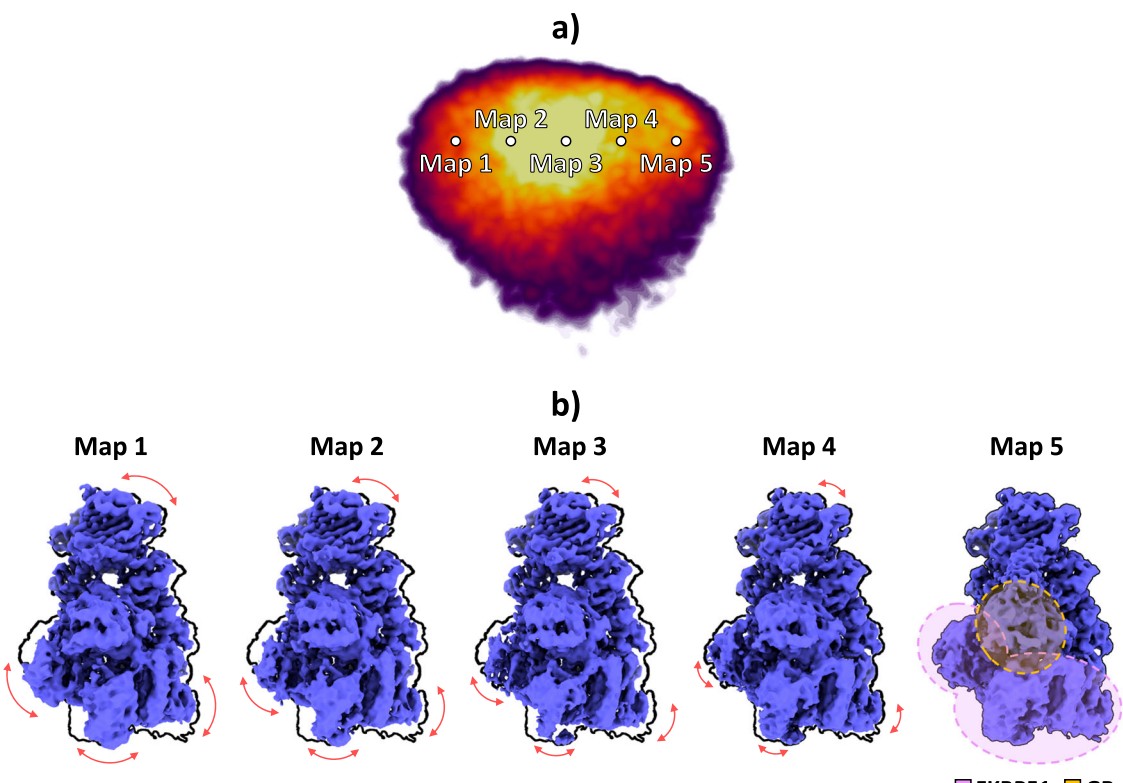

**Fig. 5 | Conformational landscape analysis of the main motions detected by HetSIREN on the GR:Hsp90:FKBP51 complex.** (**a**) shows the reduced PCA landscape obtained from the original 8D HetSIREN latent space learned by the network. Each dot in the landscape represents an even sampling along the main PC component. (**b**) displays the volumes decoded by the HetSIREN network from the sampled points shown in (**a**). The black outline shows the structural state represented by Map 5, which is provided to simplify the interpretation of the conformational change. The detected motion significantly translates the GR and FKBP51 components against the HsP90 protein.

Furthermore, we directly compared the local resolution computed by DeepRes[20] between the HetSIREN maps and the maps obtained using discrete image processing procedures, including initial model generation and subsequent refinement. This comparison revealed an increase in local resolution on the HetSIREN map, with particular emphasis on the RBD and NTD, which are typically the most mobile regions and exhibit lower resolvability (Fig. 10). To quantitatively assess the improved local resolution, we performed automatic RBD modeling with ModelAngelo[31], which showed an increase in the total number of modeled residues of 3.7 to 10.5% depending on the map (Supplementary Table 2).

By analyzing the volumes presented in Fig. 10, it is also possible to assess the reliability and precision of the method compared to discrete approaches. To that end, we compared the reconstruction obtained by CryoSparc using the 23k closest particles to the selected state as input against the map decoded by the network at the centroid of this subset in latent space. This comparison is a reasonable way to validate the method's accuracy in identifying the structural states captured by the experimental particles, avoiding hallucinations. As seen in Fig. 10, both HetSIREN and the reconstructed map agree with high confidence about the conformational state captured in that specific region of the structural landscape. In addition, the comparison of the different local resolutions estimated by DeepRes also shows the improved volume representation capabilities of the network, which is capable of decoding high-resolution states directly from a single point in the latent space, helping in those common cases in which several rounds of discrete classification may end up with reduced data sets. This capability improved the average local resolution of around 0.7 Å.

Our findings suggest that when the Spike protein is maintained at 4 °C, it tends to adopt more open configurations, predisposing it to subsequent denaturation, consistent with previous studies[26,27] at different conditions. Conversely, at 37 °C, the range of molecular motions tends toward a more closed and less accessible conformation. Even when the Spike protein is in a 1 Up state at this temperature, its opening range is markedly reduced compared to that at 4 °C. This phenomenon may directly influence the ability of the virus to evade the immune system while maintaining its capacity to initiate successful infections by modulating the equilibrium with ACE2[32]. Given the importance of the Spike protein in vaccine development, which typically involves recombinant expression of attenuated versions stored at low temperatures, our study underscores the importance of further structural analyses with methods such as HetSIREN. A deeper understanding of the conformational dynamics under specific conditions could have profound implications for the design of new vaccine formulations[33].

## Discussion

Continuous heterogeneity is a significant breakthrough in the CryoEM field, as shown by its increasing popularity and successful applications to better understand macromolecular conformational rearrangements through experimental CryoEM data[4,5,7–12].

In this regard, we have introduced a deep learning-based heterogeneous reconstruction and refinement method called HetSIREN. HetSIREN addresses the conformational variability problem in real space by encoding particle images into a latent space based on their specific structural state, followed by a decoder capable of translating the latent space into high-resolution volumes.

HetSIREN presents several critical innovations that set it apart to all current methods. In a nutshell, by working entirely in real space, HetSIREN has been able to use and even modify altogether (in the field

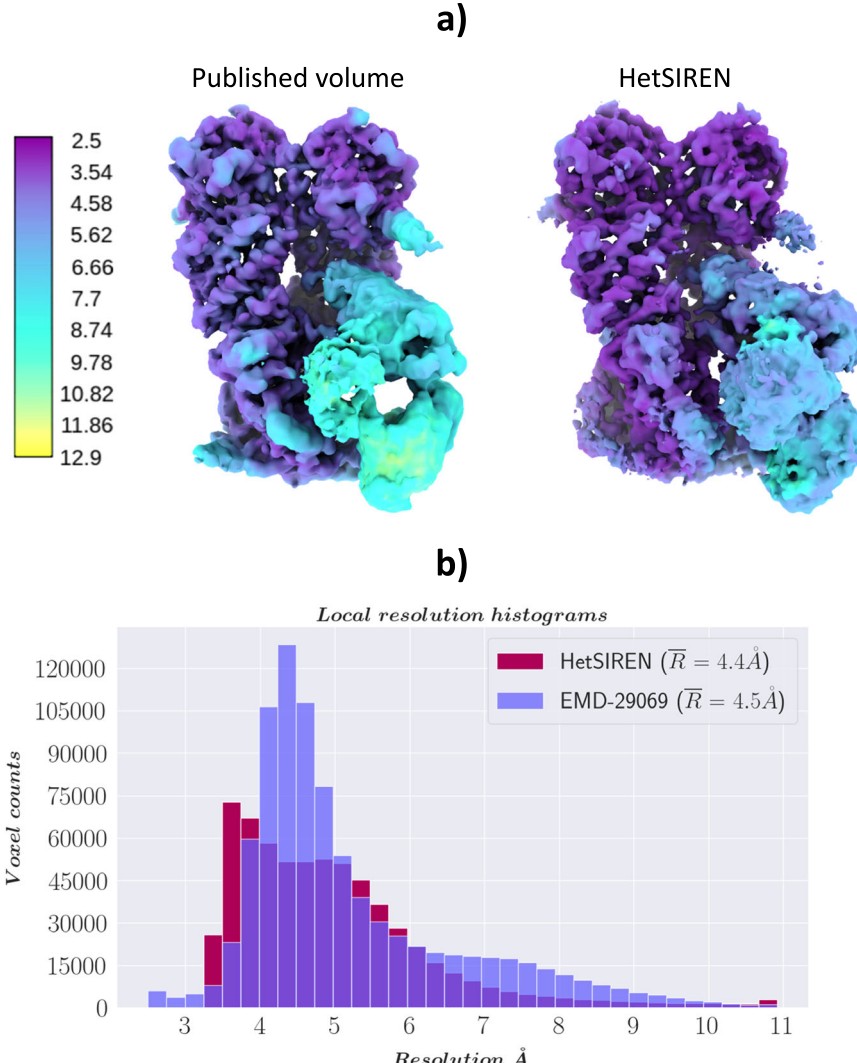

**Fig. 6 | Resolution analysis of HetSIREN decoded maps compared with the deposited map from[21].** (**a**) shows on the left the deposited map (EMD-29069) and the HetSIREN map on the right, both colored by their local resolution estimated with DeepRes[20]. The comparison shows an improvement in the local resolution of the map decoded by HetSIREN, mainly occurring in the flexible region composed of GR and FKBP51. (**b**) quantitatively compares the estimated local resolutions based on local resolution histograms. Similarly to (**a**), the local resolution of HetSIREN shows a displacement of the voxel resolutions to the high-resolution domain. Source data are provided as a Source Data file.

of CryoEM) meta-sinusoidal activation fields with many enhanced analytics capabilities to current approaches[13,34]. Furthermore, we implemented a "disentanglement" procedure concerning pose and CTF so that we introduce a constraint in latent space that makes it focus on structural differences and not into pose-induced or CTF-induced differences (indeed, the way structural changes translate into changes at the image projection level is very different depending on the particle projection direction, as indicated in ref. [14]). HetSIREN also introduces a range of regularizers, such as $L_1$ and Total Variation minimization.

The real-space decoded map is further analyzed to enhance the biomolecule signal and minimize the presence of common artifacts and errors in CryoEM, such as noise or negative values in the reconstruction, while preserving the structural features in the map. Furthermore, HetSIREN allows one to customize the reconstruction region with a mask, which can be applied to exclude unwanted structures from the decoded volumes (such as membranes or nanodiscs) or to focus the heterogeneity analysis on a specific region of space.

In addition to estimating structural states from initially supplied projection geometry information, HetSIREN can refine the initial per-particle pose and in-plane shift according to each image's specific conformation. To this end, it produces two extra latent spaces, one to analyze particle configurations and the other to refine the pose and in-plane shift of the input particles into the network. In this way, the alignment matrix refinement is also considered during the training phase, helping to generate better CryoEM maps from the decoder.

In conclusion, HetSIREN adds a new approach to the growing heterogeneity analysis family of methods. It does so by introducing unique characteristics that set it apart, including the application of SIREN-based hypernetworks to improve the quality of the decoded maps, the ability to disentangle the pose and CTF information in conformational landscapes, the possibility to customize the analysis process with user-defined reconstruction masks as well as the inclusion of explicit regularization terms to enhance the structural features of the decoded maps while reducing noise and other artifacts. The net result of all these innovations is two-fold: First, HetSIREN conformational landscapes are much more structure-focused than with current approaches in the field, and second, HetSIREN is capable of obtaining maps from individual points of the landscape with improved resolution compared to what is currently achievable in the field.

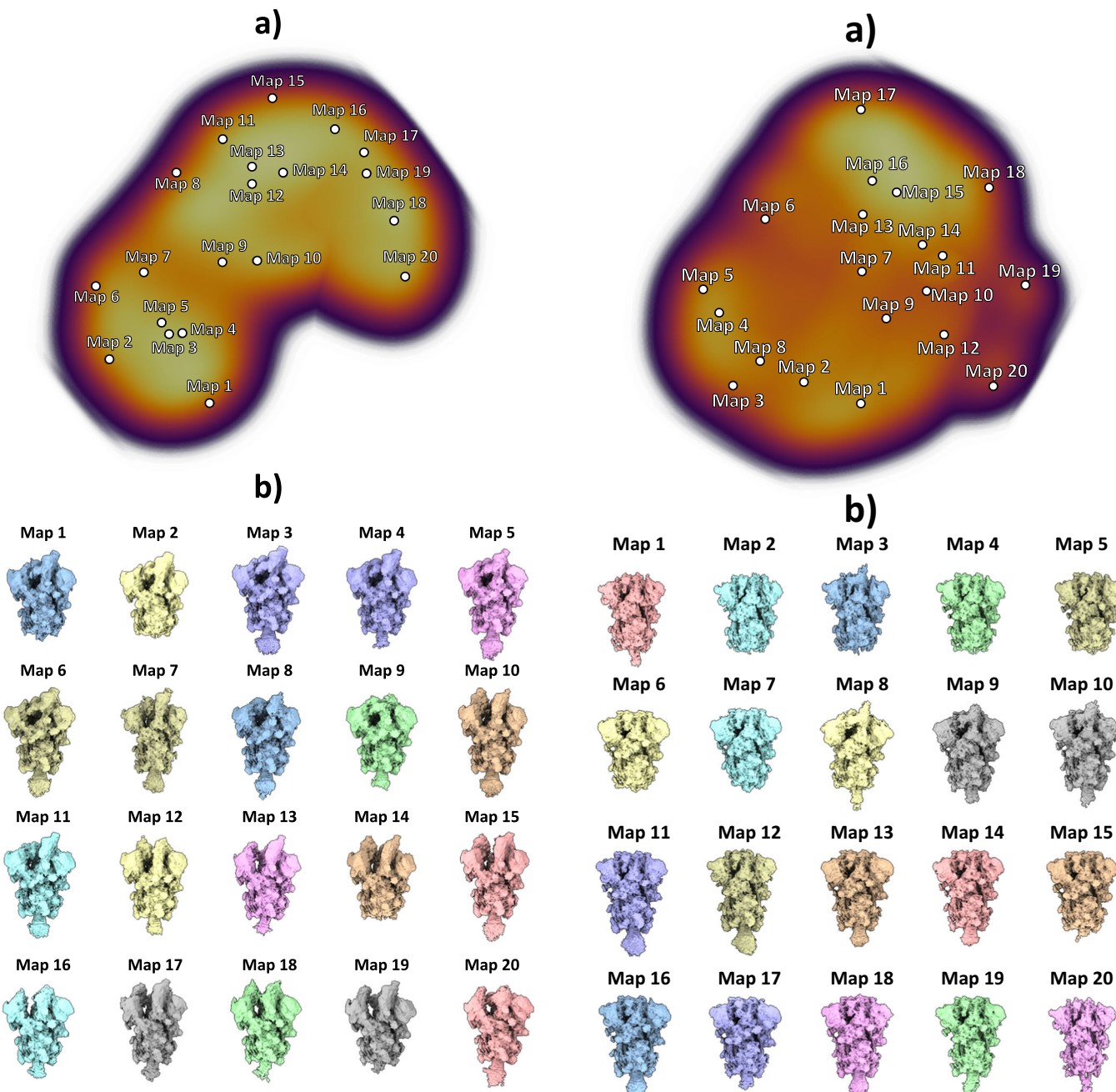

**Fig. 7 | Conformational landscape analysis of the main motions detected by HetSIREN on the SARS-CoV-2 Spike protein at 4°C.** Panel a) shows the UMAP[19] representation of the landscape obtained from the original 8D latent space encoded by HetSIREN from the particle images after training. Each dot in the landscape corresponds to the position of the cluster representative obtained from a KMeans clustering of the original HetSIREN latent space. Panel b) shows side views of the decoded HetSIREN volumes obtained from the latent space coefficients assigned to every representative shown in Panel a). The maps provide a sensible exploration of the different states identified by HetSIREN, including 1 Up and 2 Up conformations.

**Fig. 8 | Conformational landscape analysis of the main motions detected by HetSIREN on the SARS-CoV-2 Spike protein at 37°C.** Panel a) shows the UMAP[19] representation of the landscape obtained from the original 8D latent space encoded by HetSIREN from the particle images after training. Each dot in the landscape corresponds to the position of the cluster representative obtained from a KMeans clustering of the original HetSIREN latent space. Panel b) shows side views of the decoded HetSIREN volumes obtained from the latent space coefficients assigned to every representative shown in Panel a). The maps provide a sensible exploration of the different states identified by HetSIREN, including 3 Down and 1 Up conformations.

## Methods

This section starts with a general presentation of the image formation model in CryoEM and then details the architecture of the HetSIREN network and training strategies.

In addition, Supplementary Table 1 summarizes the performance analysis of the proposed method in terms of the usage of computing resources. Performance metrics were evaluated with default parameters on an RTX Ada 6000 generation GPU.

### Image formation model in CryoEM

One of the main goals of single particle analysis is to determine the 3D structure of a biomolecule through a set of 2D images generated by orthogonally integrating the electrostatic potential during micrograph acquisition. Therefore, the image formation model can be represented as a rotation operator, $R_n$, a translation operator, $T_n$, and a posterior projection, $P$, of the underlying 3D structure $V_n$. The subindex $n$

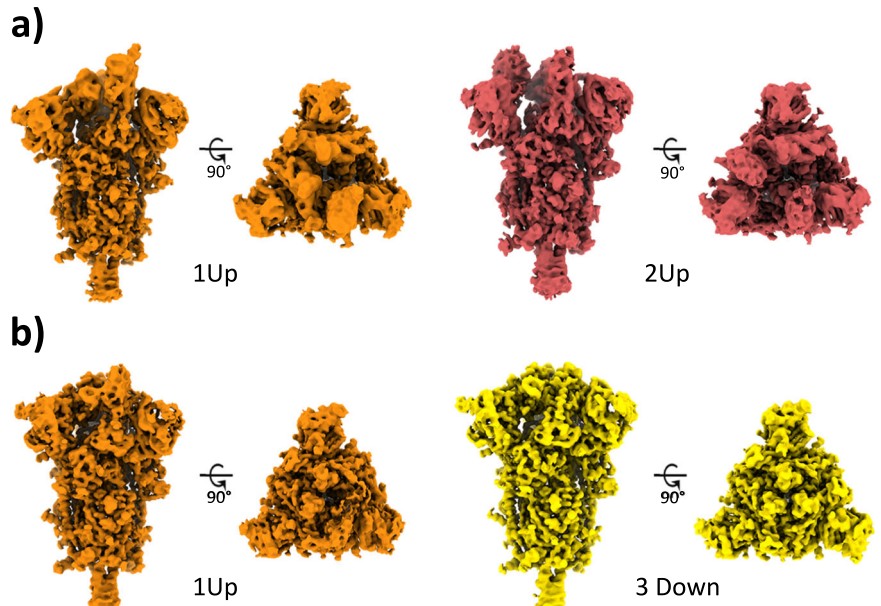

**Fig. 9 | Structural differences of the SARS-CoV-2 Spike protein at different temperatures.** Representative maps of the main conformational states detected by the HetSIREN network when the Spike protein is kept at 4°C (**a**) or 37°C (**b**). At 4°C (**a**) the Spike always shows at least one (orange) or two (red) RBDs in its open position. At 37°C (**b**) the Spike is mainly detected in its 3 Down conformation (yellow), with a lower population of particles exhibiting a 1 Up state (orange).

emphasizes that each image has a different rotation, translation, and underlying structure.

Although it is common practice to formulate the image formation model in Fourier space to take advantage of the central slice theorem, HetSIREN directly decodes the 3D structure $V_n$ in real space. Equation 1 shows the formulation of the image formation model in real space as used in this work.

$$I_n = PSF_n * (P \circ T_n \circ R_n)(V_n) + \epsilon_n \qquad (1)$$

where $\epsilon_n$ is a term representing the noise added to the image, $PSF_n$ is the point spread function that captures the optical characteristics of the CryoEM microscope, and * is the convolution operator.

## Sinusoidal Representation Network (SIREN)

HetSIREN is based on a (modified) Sinusoidal Architecture Network (SIREN)[13] to increase the quality of the high-frequency characteristics of the decoded 3D structures.

The main contribution of SIREN architectures is using sinusoidal functions as activation functions of the neural network[13]. This activation has been shown to have faster convergence and higher representation fidelity than other popular representations of signals in deep learning, such as ReLU with positional encoding, traditionally used in CryoEM[6]. In addition, the gradient computation in a sine activation can be easily modulated, as it is represented by another SIREN activation function with a phase shift, which allows for finer tuning of the gradient computations to improve the representation capabilities of neural networks.

As described in the next section, the decoder architecture in HetSIREN relies on the real-space representation of 3D signals modulated by a set of SIREN activations in its hidden layers. A dense layer with linear activation follows this to compute the map values.

Whereas SIREN-based networks have improved signal approximation capabilities in many applications[13], they suffer from a decrease in performance when representing a whole set of different structures[35], which is a critical requirement in an application such as heterogeneity analysis. Consequently, the success of SIREN in this application has

required the development of a modified architecture so that traditional meta-sinusoidal representations rely on two different layers sharing their weights to improve the inpainting capabilities on whole datasets (Supplementary fig. 3). In this meta-representation, one of the layers (commonly represented by a dense representation with ReLU activation) is used to compute the weights that will be posteriorly passed to the second SIREN layer to perform the forward pass through the network. During our experiments, we found the meta-architecture to have improved performance at the expense of slightly higher memory consumption.

It should be noted that it is possible to increase the number of ReLU layers used to compute the shared weights of its associated meta-sinusoidal layer to increase the inpainting capabilities of the model at the expense of more restricted time and GPU memory constraints.

## HetSIREN network architecture

HetSIREN network follows an autoencoder architecture detailed in Supplementary fig. 4.

The proposed encoder implements two different architectures that lead to a latent space of a number of dimensions determined by the user and set by default to 10. The architectures implemented in the encoder include a multilayer perceptron network (MLP) with three hidden layers of 1024 neurons or a residual convolutional architecture. In practice, the two encoder architectures have obtained similar latent-space representations. However, MLPs have a higher chance of overfitting, while the convolutional architecture is more robust at the expense of slightly longer training times. By default, the convolutional architecture is chosen (Supplementary fig. 5), although it can be modified in the Scipion protocol to use the MLP model if desired.

The feature vectors extracted from the latent space $z$ are then sent through the decoder, which performs the mapping $\Delta V_n = f(z_n)$ (i.e., the feature vector of the $n$-th particle is mapped to a specific map). The density values $\Delta V_n$ are then added to a reference map $V_0$ to obtain the final heterogeneous reconstruction:

$$V_n = V_0 + \Delta V_n \qquad (2)$$

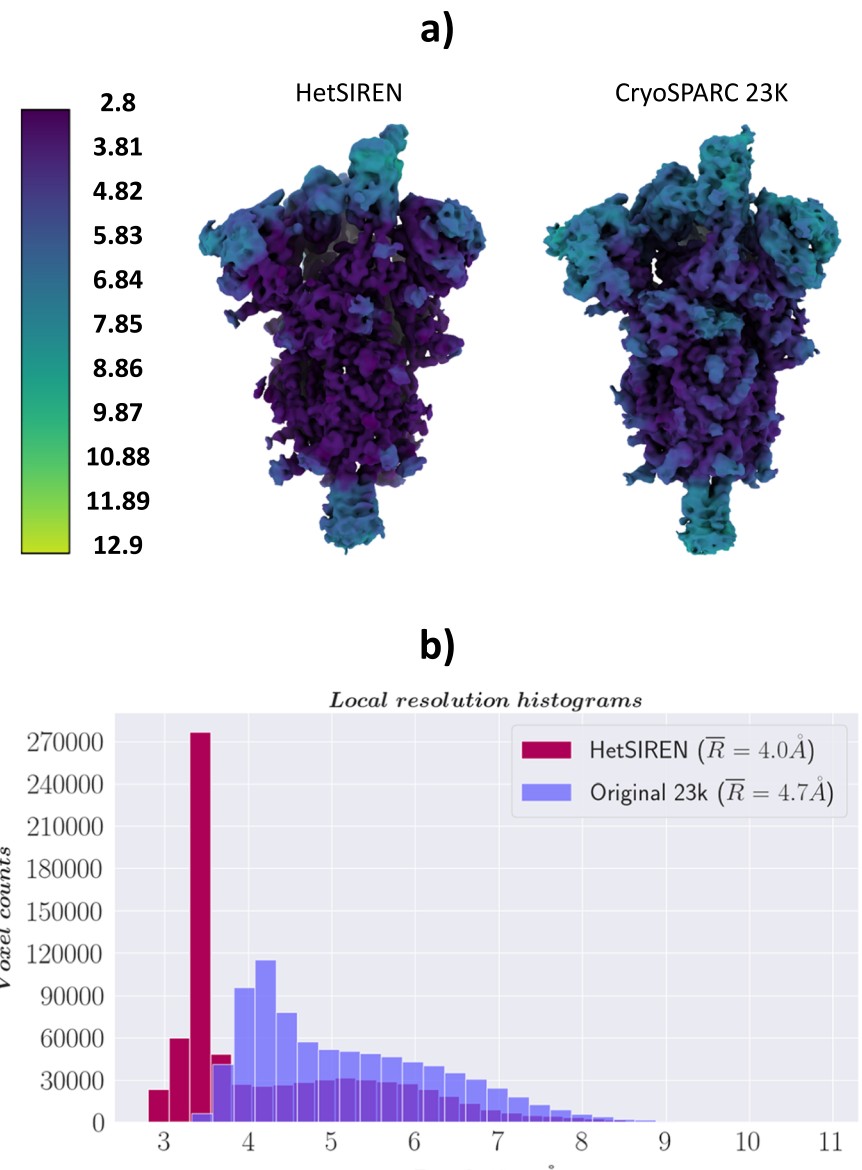

**Fig. 10 | Resolution analysis of HetSIREN compared against the map refined with standard procedures. (a)** shows the HetSIREN decoded map for one of the 1 Up conformation clusters (left), followed by the CryoSparc reconstruction of the 23766 particles closest to that cluster (right). Maps are colored according to their local resolution estimated with DeepRes[20]. **(b)** Quantitatively compares the estimated local resolutions based on local resolution histograms. Similarly to (**a**), the local resolution of HetSIREN (dark red) shows a displacement of the voxel resolutions to the high-resolution domain, translating into an improvement of 0.7 Å in the mean resolution to the CryoSparc reconstruction with the 23776 particles (light purple). Source data are provided as a Source Data file.

If the reference volume is empty $V_0 = 0$, the decoder will directly generate the volume representing a given conformation $V_n = \Delta V_n$. Suppose that the reference volume is a homogeneously reconstructed density map. In that case, the network will produce the changes that will be applied to the reference to represent a new conformational state as shown in Eq. 2.

As described in the previous section, the decoder comprises a series of hidden meta-sinusoidal residual layers followed by a dense layer with linear activation that recovers $\Delta V_n$. To keep the memory footprint of the decoder as low as possible, the number of hidden layers is fixed to three, with a total number of neurons and hyperneurons (i.e., the number of neurons in the dense ReLU layers composing the meta-sinusoidal layers) equal to the latent space dimension. The last dense layer has as many features as the density values needed to recover $\Delta V_n$.

The previous decoder is followed by a physics-based decoder that implements the image formation model defined in Eq. 1. The generated computer-simulated projections are then compared with the experimental images to backpropagate the final loss during the training phase.

## Disetangling of poses and CTF from conformations

Ideally, conformational latent spaces should only capture information on the structural changes a biomolecule may undergo based on the experimental data collected. However, conformational latent spaces suffer from a strong coupling of several factors apart from structural information, such as image pose and (to a lesser extent) CTF information. Indeed, in how structural changes are translated into projection images, differences depend strongly on the particle's projection geometry (the pose). Therefore, the interpretability of the estimated

landscapes is largely compromised unless the effect of the previous factors is explicitly decoupled from the conformational landscapes.

Following a similar process to the one conceptually proposed in ref. 14, HetSIREN includes a decoupling architecture to effectively disentangle pose and CTF from the structural information captured in its conformational landscape. By coupling the disentangled landscapes with the high-resolution volume decoder, HetSIREN allows us to explore conformational landscapes with an improved understanding of the structural features that participate in different motions.

Supplementary fig. 6 shows the modified encoder architecture, including the decoupling workflow for both the poses and the CTF information. During the training phase, experimental images are forwarded through the encoder and the decoder, generating a cleaned set of theoretical projections, the CTF corrupted theoretical projections, and the latent space vectors $\boldsymbol{z}$.

The pose decoupling step relies on a second forward pass through the decoder to generate a new set of cleaned projections. Before that, the pose information associated with the current batch of images is shuffled and passed to the decoder with the corresponding latent space vectors encoded from the experimental images. Therefore, the second forward pass will generate a new set of cleaned images from the same conformation as the first generated projections but with a different pose. A random pose could also be passed to the decoder for this step, but shuffling ensures that the original pose distribution is maintained. The two sets of cleaned images are then forwarded through the pose decoupling encoder and the conformational latent layer to generate two new sets of latent vectors $\boldsymbol{z}_t$ and $\boldsymbol{z}_{p,t}$. Since these new latent vectors encode the same conformation as the first-generated vectors but at different poses, we can impose a constraint to place them as close as possible in the latent space:

$$\text{Loss} = \lambda_p \left( |\boldsymbol{z} - \boldsymbol{z}_t|_2^2 + |\boldsymbol{z} - \boldsymbol{z}_{p,t}|_2^2 \right) \tag{3}$$

Upon convergence, the network will learn to produce the same conformation independently of the image pose, effectively decoupling the structural and pose information.

The CTF decoupling process follows a principle similar to the pose decoupling workflow described above. In this case, apart from the CTF corrupted theoretical projections, a new set of CTF corrupted images is generated through a third forward pass through the decoder. Before this new forward pass, the CTFs associated with the batch are shuffled without touching the poses. The new sets of images are then passed to the CTF decoupling encoder and the conformational latent layer to generate two new sets of latent vectors $\boldsymbol{z}_c$ and $\boldsymbol{z}_{p,c}$. Similarly to the previous case, the new latent vectors and the first generated vectors should be as close as possible in the latent space, as they represent the same conformation up to the CTF corruption. Therefore, we can impose a new regularization factor as follows:

$$\text{Loss} = \lambda_c \left( |\boldsymbol{z} - \boldsymbol{z}_c|_2^2 + |\boldsymbol{z} - \boldsymbol{z}_{p,c}|_2^2 \right) \tag{4}$$

Once convergence is achieved, the network will learn to produce the same conformation independently of the CTF, effectively decoupling the structural and CTF information.

In the last instance, combining the two decoupling workflows allows the generation of a decoupled latent space where the conformational information predominates, increasing the interpretability of the conformational landscape.

## HetSIREN cost function
The possibility of working directly in real space when decoding theoretical volumes/images allows the inclusion of additional constraints in the training objective function. These extra terms prevent the network from learning unwanted or meaningless features in the experimental images, such as noise or normalization errors.

Before introducing the different terms included in the HetSIREN objective function, we provide a simplified guideline of the implemented training strategy. The optimization of network parameters is based on the Adam optimization method with a custom learning rate (set by default at $10^{-5}$) and a batch size (set by default to 8). Experimental images are forwarded through the network, leading the network output to a set of theoretically decoded projections. The comparison of experimental and theoretical projections uses, by default, a standard Mean Square Error (MSE) function as follows:

$$\text{Loss} = \sum_b |I_b - D(\boldsymbol{z}_b)|_2^2 \tag{5}$$

where $I_b$ is an experimental image in the batch of images being considered, $D$ represents the decoder network, and $z_b$ is the latent space vector associated with $I_b$ by the encoder network:

$$\boldsymbol{z}_b = E(I_b) \tag{6}$$

Besides the standard MSE cost function, we provide alternative means to compare the theoretical and experimental projections to further customize the network training in the Scipion protocol, such as the correlation between the image pair.

## Real-space regularization
The low signal-to-noise ratios encountered in particle images extracted from the acquired micrographs are probably the primary source of errors and overfitting in CryoEM image processing.

In a homogeneous reconstruction, one can use averaging of many images during the reconstruction process to reduce the noise level as much as possible. Nevertheless, the previous solution does not apply to the heterogeneous reconstruction case, where, ideally, the reconstruction of 3D structures per particle is the primary goal.

One possible way to regularize the noise in the decoded volumes is to apply a low-pass filter in the reconstructed Fourier space at the expense of decreasing the resolution of the reconstructed structures, which we tend to avoid. Fortunately, noise can be regularized in real space without sacrificing the high-frequency information content discarded by the low-pass filter, as indicated in the next paragraph.

Returning to the image formation model described in Eq. 1, it is possible to observe that the effect of the noise term $\epsilon$ is the addition of additional unwanted density values to the voxels in the volume/image grid. Therefore, we may penalize the cost function with an $L_1$ regularization term:

$$\text{Loss} = \ldots + \lambda_1 |\boldsymbol{V}_0 + \Delta \boldsymbol{V}_n|_1 \tag{7}$$

The previous term enforces HetSIREN to learn a $\Delta \boldsymbol{V}_n$ that minimizes noise while preserving the structural high-frequency features of the decoded volumes. Depending on the conditions of the data set and the desired denoising level, it is possible to modify the regularization term $\lambda_1$ to improve the quality of the decoded structures. By default, the regularization parameter is set to 1.0, which we have practically found to introduce a good balance between the loss function terms for all the datasets currently tested.

## Negative value mitigation
Ensuring proper background and noise normalization in CryoEM images introduces an artifact in the reconstructed structures represented as a set of negative values scattered along the volume grid. Although this is not usually a significant concern, we found that heterogeneous reconstruction benefits from regularizing this artifact,

allowing the neural network to focus on the protein signal instead of compensating the generated values with unwanted negative voxels.

Therefore, the objective function is further modified, including a $L_1$ regularization term that penalizes adding negative values to the volume.

$$\text{Loss} = \ldots + \lambda_2 |\min(\boldsymbol{V}_0 + \Delta\boldsymbol{V}_n, 0)|_1 \qquad (8)$$

Similarly to the previous case, controlling the regularization strength through the regularization parameter $\lambda_2$ is possible, which is set by default to 1.0.

### Enforcing density continuity in decoded maps

Even in the presence of the denoising regularization term, overfitting might remain an issue, especially when working with large-volume grids to achieve high-resolution 3D structures. However, the nature of the overfitting differs from the addition of noise introduced previously: The decoded voxel values might appear as artifacts scattered along the protein signal so that the projection still matches the proper structure but without providing meaningful structural features at the volume level.

Therefore, ensuring proper density continuity is essential to allow the network to learn high-frequency structural details while maintaining a proper biomolecular shape. Our model controls density continuity through Total Variation (TV) losses, which account for different continuity features. The rationale behind TV is to promote overall smoothness in the image by reducing noise and minor fluctuations while allowing for sharper edges. Combining both penalizations encourages the decoded volume to have smooth transitions with a reduced likelihood of abrupt changes in pixel values while preserving edges that might otherwise be overly smooth.

Our implementation of TV regularizations is:

$$\text{Loss} = \ldots + \lambda_3 |\nabla(\boldsymbol{V}_0 + \Delta\boldsymbol{V}_n)|_1 + \lambda_4 |\nabla(\boldsymbol{V}_0 + \Delta\boldsymbol{V}_n)|_2^2 \qquad (9)$$

where $\nabla\boldsymbol{V}$ represents the spatial gradient of $\boldsymbol{V}$.

As in previous regularization terms, the regularization strength can be controlled through the parameters $\lambda_3$ and $\lambda_4$, respectively; by default, both terms are set to 0.1.

### Multiresolution loss to achieve high resolution in a single training

The two main objectives of heterogeneous reconstructions are to provide a meaningful latent space that orders the conformational states captured by the particles according to their similarity and to decode high-resolution maps with the decoder to explore and explain the latent space. The previous workflow generally involves training the network with the original images at full resolution. However, in practice, the previous approach is not always ensured to converge to a satisfactory solution due to the large amount of local minima present when minimizing the objective function.

One possible approach to overcome the local minima problem is to warm up the neural network. This implies initial training on downsampled images, which smooths the solution space, thus minimizing the chances of getting stuck on spurious local minima. The pretrained network is then fine-tuned on the unsampled data, allowing it to reach high-resolution structures without escaping from the initial solution.

However, the previous approximation requires at least two different training steps, which overall impacts the learning time needed by the network. In HetSIREN, we propose a multiresolution training approach that allows for robustly obtaining a high-resolution structure on a single model training step, which implies a significant improvement in the training time compared to the previously described strategy. The multiresolution approach minimizes the MSE between different pairs of images at different resolutions, allowing the network

to explore both the smooth solution space defined by the filtered image pairs and the original solution space. In this way, the loss function becomes:

$$\text{Loss} = \sum_b \sum_\omega |L_\omega(I_b - D(\boldsymbol{z}_b))|_2^2 \qquad (10)$$

where $L_\omega$ is a lowpass filter followed by a downsampling operator and $\omega$ is chosen from a discrete set of cutoff frequencies.

During multiresolution training, the original experimental images are forwarded through the network and used to decode volumes at the same pixel size as the experimental images. A bank of filters and downsampling operations is posteriorly applied to the experimental and decoded projections to generate the multi-resolution pairs. Finally, the pairs are compared through an MSE error loss and combined before backpropagation occurs.

It should be noted that the previously described regularization terms are only computed with the original decoded map at full resolution. In this way, the network focuses on improving the features of the unsampled volumes at the original pixel size.

In our tests, we discovered that using only a set of frequencies at full resolution ($R$) and half resolution ($2R$) typically achieves the desired outcomes, streamlining the trade-off between the MSE costs of the original and downsampled image pairs.

### Focused reconstruction in HetSIREN

On many occasions, 3D structure reconstruction in CryoEM is carried out on the entire volume grid that contains the biomolecule of interest (or, for memory-saving purposes, on the sphere inscribed in the cubic grid). However, the region of interest might be more complex in some scenarios. For example, nanodiscs or membranes in the reconstructed maps are usually undesired as they might affect the proper reconstruction of the embedded structural features.

In heterogeneous reconstruction, the motivation for designing such a mask follows a similar reasoning, with the addition of focusing the latent space so that only the conformational changes in a region of interest are captured. Therefore, HetSIREN allows custom reconstruction masks that determine the area to be reconstructed by the neural network.

The implementation of focused reconstruction in HetSIREN is based on a mask $M$ designed and input by the user in the form of the Scipion protocol. The network configuration is modified to accommodate the focused reconstruction process if a mask is provided. The main change applied to the network is to limit the number of voxels considered in the volume decoder $D$ to only those present in the mask as follows:

$$\Delta\boldsymbol{V}_n = D_M(\boldsymbol{z}) \qquad (11)$$

Here, $D_M$ represents the new decoder focused on mask voxels. The previous modification allows us to generate theoretical projections only for the structural changes within the mask.

Even though the network will learn to modify only the regions within the mask, we found it helpful to consider the voxels out of these regions when generating the theoretical 2D projections. Thanks to the $\Delta\boldsymbol{V}$ implementation in HetSIREN, it is possible to project the entire 3D volume along a given particle projection direction once the desired reconstruction region has been refined according to the decoded values. In this way, obtaining a set of theoretical projections with homogeneous information is possible apart from the area enclosed by the mask defined by the user.

Once the 2D projections have been generated, it is possible to use the cost function and regularization previously described to train the network. However, the previous cost functions will not ensure that the region being refined/reconstructed will follow a similar voxel value

distribution to the one in the original map. Therefore, when focusing on the landscape, an additional regularization parameter is added to ensure that the application of $\Delta V$ respects the voxel value distribution of the reference volume. Being $\boldsymbol{v}^{D,n}$ and $\boldsymbol{v}^{D,r}$ the vector storing the voxel values within the region of interest for the HetSIREN and reference volumes, respectively, the new regularization reads:

$$
\begin{aligned}
Loss = \ldots + \lambda_5 \Bigg( & \sum_b \left( \max(\boldsymbol{v}_b^{D,n}) - \max(\boldsymbol{v}_b^{D,r}) \right)^2 \\
& + \sum_b \left( \min(\boldsymbol{v}_b^{D,n}) - \min(\boldsymbol{v}_b^{D,r}) \right)^2 \\
& + \sum_b \left( \mu_b^{D,n} - \mu_b^{D,r} \right)^2 \\
& + \sum_b \left( \sigma_b^{D,n} - \sigma_b^{D,r} \right)^2 \Bigg)
\end{aligned}
\tag{12}
$$

where $\mu$ and $\sigma$ represent the mean and standard deviation values stored in the vectors.

### Reporting summary

Further information on research design is available in the Nature Portfolio Reporting Summary linked to this article.

## Data availability

The atomic coordinates and cryo-EM density maps for the SARS-CoV-2 Spike protein at 4 °C and 37 °C were deposited in the Protein Data Bank and EM Data Bank with codes 9GDX and 9GDY and EMD-51279 and EMD-51280, respectively. The synthetic and ribosome dataset analyzed in this work can be downloaded as a Scipion test dataset through the following command: scipion3 testdata --download FlexHub_Tutorials (assuming Scipion is already installed in the system). The atomic model used in the synthetic dataset is deposited in the Protein Data Bank with code 4AKE. The source data underlying Figs. 4b, 6b, 10b and Supplementary fig. 7a, b, d are provided as a Source Data file. Source data are provided with this paper.

## Code availability

HetSIREN algorithm is freely available through Scipion 3.0[16] under the plugin scipion-em-flexutils[36] https://github.com/scipion-em/scipion-em-flexutils and the package Flexutils-Toolkit[37] https://github.com/I2PC/Flexutils-Toolkit. The protocol corresponding to the algorithm described in this manuscript is flexutils - flexible align - HetSIREN. Tutorials on how to setup and use HetSIREN are provided in the following webpage https://scipion-em.github.io/docs/release-3.0.0/docs/user/tutorials/flexibilityHub/main_page.html#tutorials.

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

## Acknowledgements

The authors acknowledge the financial support from the Ministry of Science, Innovation and Universities (BDNS n. 716450) to the Instruct Image Processing Center (I2PC) as part of the Spanish participation in Instruct-ERIC, the European Strategic Infrastructure Project (ESFRI) in the area of Structural Biology, Grant [PID2022-136594NB-I00] (COSS and JMC) funded by MICIU /AEI/10.13039/501100011033/ and "ERDF A way of making Europe", by the "European Union", Spanish State Research Agency AEI/10.13039/501100011033, through the "Severo Ochoa" Programme for Centres of Excellence in R\&D [CEX2023-001386-S] (COSS and JMC), Comunidad Autónoma de Madrid" through Grant: S2022/BMD-7232 (COSS and JMC), European Union (EU) and Horizon 2020 through grant: HighResCells (ERC - 2018 - SyG, Proposal: 810057) (COSS and JMC) European Union (EU), and Horizon Europe through grant: Fragment Screen Proposal: 101094131 (COSS and JMC). We thank Medigen Vaccine Biologics Corp. (MVC, No. 68, Shengyi 3rd Rd., Zhubei City, Hsinchu County 302, Taiwan) for providing the SARS-CoV-2 spike protein used in this work. The SARS-CoV-2 spike protein cryo-EM work was supported by Academia Sinica (AS-KPQ-109-TPP2) (MDT) and Taiwan Cryo-EM Consortium (grant no. NSTC 113-2740-B-006-004) (MDT). Cryo-EM experiments were performed at the Academia Sinica Cryo-EM Facility supported by Academia Sinica (AS-CFII-108-110) (MDT), and initial image processing steps were performed at the Academia Sinica Grid-computing Center. We also thank Dr. Mohamad Harastani for helping generate the adenylate kinase dataset.

## Author contributions

D.H. developed and tested the HetSIREN method presented throughout the manuscript. C.P.M. and J.K. processed and analyzed the datasets presented in the manuscript. D.I. prepared cryo-EM grids, collected data, and pre-processed the SARS-CoV-2 datasets. C.N. and DA provided and helped to understand the GR:Hsp90:FKBP51 dataset. M.D.T. provided and helped with the understanding of the SARS-CoV-2 Spike datasets. C.O.S.S. and J.M.C. jointly supervised this work. D.H. and C.P.M. wrote the manuscript.

## Competing interests

The authors declare no competing interests.
