## [Transparent Peer Review file · Nature Communications]

Real-space heterogeneous reconstruction, refinement, and disentanglement of CryoEM conformational states with HetSIREN

Corresponding Author: Dr David Herreros

Version 0:

Reviewer comments:

Reviewer #1

(Remarks to the Author)

In this manuscript, the authors present a deep neural network based algorithm that learns the structural heterogeneity of proteins from single particle CryoEM data. The method is carefully designed to disentangle the uncertainty of pose and CTF estimation from the actual structural variability, and as a real space method, it allows masking to focus on the dynamics of regions of interest. I find this paper highly informative and nicely written, which would benefit readers both in the field of deep learning methods and structural biology. My main comments listed below focus on the presentation of some of the novel features and how to better demonstrate their usefulness to the audience.

I am quite grateful to learn about the SIREN from this paper. I tested the sinusoidal activation function on some deep learning applications (unrelated to the work in the paper) myself, and it does seem to outperform ReLU slightly. That being said, I don't think most readers would spend time doing the tests. Since SIREN is one of the main selling points of the paper, it might be worth adding a comparison of that versus the traditional ReLU network, even just on a simple, simulated dataset.

For the CTF decoupling, while it is better to have conformations along a straight line in the latent space, it is not necessarily a good indication of CTF decoupling. One simple way is to add the ground truth defocus of particles in the latent space plot. If the CTF information is entangled with conformational changes in the standard method and decoupled in the modified version, in the plots the conformation would be somewhat correlated with the defocus values with the standard method, and independent in the decoupled version.

The scheme for pose decoupling is well designed. Similar to the CTF decoupling, I would recommend demonstrating the performance of the pose decoupling module by itself. For example, simply add noise to the assigned orientation of input particles before the heterogeneity analysis. If everything works as expected, the standard method would get much worse results, and the pose decoupled version should produce similar results with or without noise in the pose assignment.

While the resolution improvement shown in figure 4 is convincing, it is less impressive in figure 6. It seems that some of the flexible parts get better resolved, but the rigid parts also get more hollow. It would be good to use a comparable threshold for the isosurface rendering, and also add some zoomed-in views to show the improvement of the high resolution features.

For Figure 7 and 8, is there a good reason to arrange the maps by that order? It would be easier to visualize the conformational changes if the authors sort the maps by their locations in the latent space.

(Remarks on code availability)

I spent some time on this but failed to find the actual implementation of the algorithm in the GitHub repository. The repository provide a wrapper for the HetSiren protocol, but the actual code of that seems to be somewhere else. It would be good if the authors provide a link to the code.

Reviewer #2

(Remarks to the Author)

This paper presents the HetSIREN method for effectively achieving multi-conformational 3D reconstruction in cryo-EM. The authors have conducted extensive experiments on various datasets, including simulated and experimental images, to demonstrate the method's effectiveness. They also highlight that HetSIREN can better disentangle different conformations in the latent space by optimizing pose and CTF, while achieving denoising for higher-resolution reconstructions. However, there are still some concerns that need to be addressed.

There is a need for more detailed comparisons and experimental evidence to support its uniqueness, particularly on the new GR:Hsp90:FKBP51 dataset, where analyses using other methods (e.g., cryoDRGN) are lacking.

The authors have designed numerous complex loss functions to constrain the training process, but there is a lack of systematic discussion on whether these losses have a meaningful impact on the results. I suggest conducting ablation studies to demonstrate the effectiveness of these losses on prediction outcomes.

The authors emphasize the significant denoising effects of their method. However, the paper lacks a convincing discussion of the denoising results. It is known that density maps generally require sharpening after 3D reconstruction, or post-processing using current deep learning methods like DeepEMhancer and EMReady. These deep learning post-processing methods largely achieve denoising effects. Does the authors' claim imply that HetSIREN has integrated a full pipeline of 3D reconstruction with deep learning-based post-processing? I would be very interested to see a comparison between the reconstructions using HetSIREN and those using cryoSPARC with other methods like DeepEMhancer and/or EMReady.

(Remarks on code availability)

Version 1:

Reviewer comments:

Reviewer #1

(Remarks to the Author)

The authors have carefully addressed all my comments in the revised manuscript, and the additional experiments nicely demonstrated the performance of the individual modules of HetSIREN. I am overall happy with the revised version and recommend the publication of the paper.

Some minor comments:

In supplementary figure 9, the text in the figure says CTF decoupled while the caption talks about pose decoupling.

Consider using a brighter color scheme for Figure 6a. Currently the high resolution part of the protein is so dark the features are hardly distinguishable.

In supplementary figure 2, if the authors claim the extra density to be a sidechain, it is necessary to show the sidechain atoms of the model and have it roughly fit into the density.

(Remarks on code availability)

The code seems reasonably well-arranged. I still have not found a tutorial on how to use the method. A link to such a tutorial in the manuscript would be helpful.

Reviewer #2

(Remarks to the Author)

The authors have added detailed comparative results related to cryoDRGN and DeepEMHancer, which addressed most of my comments. It would also be valuable to give a comparison with EMReady or a discussion on the topic as it has been widely used in the cryo-EM field. Other than that, I have no more comments.

(Remarks on code availability)

RESPONSE LETTER

In the following document, we include our detailed answers to the reviewer's comments, including a description of the modifications in the manuscript to simplify their tracking in the new version. The letter is formatted so the reviewer's comments are highlighted in blue, followed by the answers in dark green. For the completeness of the response letter, we also include the original text included in the reviewed version of the manuscript in purple at the end of every answer.

Reviewer #1

Remarks to the Author

In this manuscript, the authors present a deep neural network based algorithm that learns the structural heterogeneity of proteins from single particle CryoEM data. The method is carefully designed to disentangle the uncertainty of pose and CTF estimation from the actual structural variability, and as a real space method, it allows masking to focus on the dynamics of regions of interest. I find this paper highly informative and nicely written, which would benefit readers both in the field of deep learning methods and structural biology. My main comments listed below focus on the presentation of some of the novel features and how to better demonstrate their usefulness to the audience.

I am quite grateful to learn about the SIREN from this paper. I tested the sinusoidal activation function on some deep learning applications (unrelated to the work in the paper) myself, and it does seem to outperform ReLU slightly. That being said, I don't think most readers would spend time doing the tests. Since SIREN is one of the main selling points of the paper, it might be worth adding a comparison of that versus the traditional ReLU network, even just on a simple, simulated dataset.

Following the reviewer's suggestion, we have included a new supplementary section in the manuscript titled "Comparison of SIREN and ReLU activation in HetSIREN," assessing in more detail the effect of changing the activation function in the architecture proposed in HetSIREN.

The results obtained from this analysis are summarized in Supplementary Figure 9, which is also presented below:

Supplementary Figure 9: Comparison of the decoded accuracy of HetSIREN when trained using two different activation functions in the decoder: ReLU and SIREN. The comparison shows that both activations have a similar performance in representing the structural details in a given state, although SIREN gives more freedom to the network to represent strong compositional variations, as highlighted in the upper images.

It should be noted that the two networks trained to evaluate the differences arising from changing the activation functions in the volume decoder do not include the decoupling architectures and the additional cost functions presented in the manuscript. The main reason behind the previous decision is to isolate as much as possible the effect of activation functions on the network's performance to properly evaluate them.

The comparison suggests that, in our architecture, ReLU and SIREN activations have a similar performance in terms of the structural details learned in the decoded volumes. However, there is a significant difference in the representation of strong compositional variations between states. While ReLU cannot completely remove the 40S subunit of the ribosome state shown in the upper images, SIREN yields better accuracy thanks to a clearer removal of the 40S density.

We include below the new text added to the manuscript for the completeness of the response letter:

Comparison of SIREN and ReLU activation in HetSIREN

Applying different activation functions to the outputs of the layers in a neural network may induce differences in the accuracy and performance of a neural network. In this manuscript, we propose the application of sine activation functions well known in the deep learning field as

SIRENs. Even though SIREN activation functions usually outperform other popular activations like ReLU, it is interesting to evaluate their effect on HetSIREN and its architecture. To properly assess the differences between SIREN and ReLU in HetSIREN, we propose a test with the EMPIAR 10028 dataset (3) analyzed throughout the manuscript, which will be used to train two networks: the first one consists of HetSIREN with ReLU activations without adding the decoupling and the additional cost functions proposed in this work to isolate the effect of the activation function. The second network follows the same principles as the first one, changing the activation function to the SIREN activations presented in this work.

After training the two networks, two different conformations were selected from the conformational landscapes, decoding two volumes representing two distinct compositional states found in the dataset: one of the conformations loses completely its 40S subunit, while the second has a smaller loss of mass in the 40S subunit of the ribosome. The comparison of these two states is presented in Supplementary Figure 9.

As can be seen from the decoded volumes, both ReLU and SIREN perform similarly in our network architecture regarding the structural details of the structures. However, a significant difference is highlighted in the upper images arising from the change in the activation function. While ReLU activations prevent the network from learning how to completely remove the 40S subunit of the ribosome, SIREN activations lead to a more sensible representation of this evasive state thanks to a clearer removal of the subunit.

For the CTF decoupling, while it is better to have conformations along a straight line in the latent space, it is not necessarily a good indication of CTF decoupling. One simple way is to add the ground truth defocus of particles in the latent space plot. If the CTF information is entangled with conformational changes in the standard method and decoupled in the modified version, in the plots the conformation would be somewhat correlated with the defocus values with the standard method, and independent in the decoupled version.

We thank the reviewer for suggesting the previous test, as it greatly simplifies understanding the decoupling effects on the organization of the latent spaces. Following this suggestion, we have included a new supplementary section, “Characterization of the decoupling effect on conformational landscapes,” with a more detailed assessment of the decoupling architecture. The tests focus on the simulated adenylyate kinase dataset in the manuscript, where a different

CTF corrupted each image. From the previous dataset, we train two different networks: the first follows a standard architecture without decoupling, while the second includes the CTF decoupling architecture. The results obtained are summarized in Supplementary Figure 8, which is also presented below:

Supplementary Figure 8: Assessment of the CTF decoupling architecture on the latent space learned by HetSIREN. Panels a) and b) show two latent spaces obtained by training two different networks with images with variable CTF corruption. Panel a) shows the landscape encoded by the network with no decoupling architecture. Panel b) shows the landscape encoded by the network, including only the CTF decoupling part. The colors used to represent the landscapes correspond to a clustering of the CTF of the images into three different groups to simplify the visualization of this information. The comparison of the two panels shows how the decoupling effect effectively condenses the latent space, reducing the spreading induced by the strong organization of the latent space according to the CTF of the images.

The two landscapes that the two trained networks learned were then colored according to their CTFs by clustering these values into three different groups. As the results show, when no decoupling is present, the latent space spreads into a band-like pattern that depends on the CTF. In contrast, the CTF decoupling architecture effectively condenses the landscape, reducing the spreading induced by the CTF and leading to a more accurate representation of the conformational space.

We include below the new text added to the manuscript for the completeness of the response letter:

Characterization of the decoupling effect on conformational landscapes

The decoupling architecture introduced in HetSIREN minimizes the effects that the pose and CTF have on the organization of the different images in a latent space. Ideally, a conformational latent space that considers only the structural differences in the particle images should be learned. This way, the conformational landscape of the biomolecule under study can be properly reflected.

To better reflect these effects in the conformational latent spaces, we propose two scenarios in which the pose and CTF's downstream effects dominate.

The first test case consists of analyzing the simulated adenylylase kinase dataset presented in the first section of the manuscript. During the simulation, a different CTF corruption was applied to each projection individually, trying to make the CTF as prominent as possible against the pose and the conformational variability captured in the images.

The previous images were used to train two different HetSIREN networks: the first network has a standard architecture without decoupling; in contrast, the second network includes the CTF decoupling architecture but not the pose decoupling part. Comparing the two landscapes allows one to better observe how the CTF decoupling architecture affects the latent space organization.

The results of this analysis are summarized in Supplementary Figure 8. The landscape colors represent a clustering of the different images according to their CTF information to simplify the visualization of the organization of the images based on the CTF information. Supplementary Figure 8a shows the landscape learned by the network with no decoupling architecture, leading to a significant landscape spreading to accommodate different "bands" with similar CTFs. This is a strong deviation from the gold standard landscape, which should be a straight line, as discussed in the manuscript section "Simulated adenylylase kinase landscape and landscape disentanglement." In contrast, the CTF-decoupled latent space shown in Supplementary Figure 8b shows a more condensed latent space, better reflecting the ideal latent space. By combining images with the same conformation and variable CTF information, HetSIREN effectively learns to decouple the CTF effect, minimizing its effect on the organization of the latent space and leading to a more prominent structural component.

The scheme for pose decoupling is well designed. Similar to the CTF decoupling, I would recommend demonstrating the performance of the pose decoupling module by itself. For example, simply add noise to the assigned orientation of input particles before the heterogeneity analysis. If everything works as expected, the standard method would get much worse results, and the pose decoupled version should produce similar results with or without noise in the pose assignment.

Similar to the previous answer, we have included a similar test on the effect of the pose decoupling in the new supplementary section, “Characterization of the decoupling effect on conformational landscapes.”

We propose a different simulated dataset for this test, obtained as the projection gallery of two SARS-CoV-2 Spike electron density maps in one-up and three-down conformations. The generated projections were not corrupted with a CTF, trying to keep the pose contributions in the latent space as relevant as possible. By construction, the gold-standard latent space that should be predicted by the method should be just two separated “dots” representing the two states used to generate the projections.

Two networks were trained with the previous image dataset: a standard HetSIREN network with no decoupling architecture and another network with only the pose decoupling part. The results obtained are summarized in Supplementary Figure 9, which are presented below:

Supplementary Figure 9: Assessment of the pose decoupling architecture on the latent space learned by HetSIREN. The panels show the latent spaces obtained by training two different networks with images with variable poses and no CTF corruption. Panel a) shows the landscapes obtained with the training dataset with a uniform pose distribution. Panel b) shows the landscapes obtained after predicting from the training dataset after adding noise to the original poses. The colors used to represent the landscapes correspond to a clustering of the pose of the images into four different groups to simplify the visualization of this information. The comparison of the two panels shows how the decoupling effect effectively condenses the latent space, reducing the spreading induced by the strong organization of the latent space according to the pose of the images.

We include below the new text added to the manuscript for the completeness of the response letter:

The second test relies on analyzing clean images without CTF, which allows us to better assess the pose effect on the conformational landscape. To that end, we simulated 2000 images from two SARS-CoV-2 Spike electron density maps in one-up and three-down conformations. This

simulated dataset describes a very simple conformational latent space, ideally consisting of two isolated points representing the two discrete states used to simulate the images.

Similar to the previous test, two HetSIREN networks were trained with the new image dataset: the first network has a standard architecture with no decoupling parts, while the second includes only the pose decoupling architecture to analyze its effect on the landscape. The results of this analysis are summarized in Supplementary Figure 9. Supplementary Figure 9a shows the landscapes obtained from the training dataset, colored according to clustering into four groups of the projection sphere to better visualize the pose. The non-decoupled landscape suffers from a similar effect to the CTF case, deviating from the ideal "two dots" latent space due to a strong organization induced by the pose. In contrast, the decoupled landscape presented is significantly condensed towards the ideal "two dots" representation, showing the ability of the architecture to effectively learn that images with similar conformation and different poses should be close in the latent space. A different experiment is proposed in Supplementary Figure 9b, where the previously trained networks are used to predict the landscape of a new dataset composed of the original images after applying noise to their poses. As can be seen from this result, the non-decoupled network predicts a disordered landscape, placing the particles in completely different locations compared to the landscape shown in panel a), even if the images are the same. In contrast, the decoupled landscape is not so much affected by the new poses, as it has learned to predict that the images represent two distinct conformations independently of their pose.

While the resolution improvement shown in figure 4 is convincing, it is less impressive in figure 6. It seems that some of the flexible parts get better resolved, but the rigid parts also get more hollow. It would be good to use a comparable threshold for the isosurface rendering, and also add some zoomed-in views to show the improvement of the high resolution features.

We have included in the manuscript a new supplementary figure (Supplementary Figure 2) with several zoom regions to make it easier to compare HetSIREN against the deposited volume for the GR:Hsp90:FKBP51 dataset. We include this new figure below to complete the letter.

Supplementary Figure 2: Detailed comparison of HetSIREN and the deposited map from (5). The different panels present several zoom regions of the two volumes to better compare the resolution changes between HetSIREN and the deposited map. In addition, we highlight in the last row how HetSIREN has the ability to detect small structural details like side chains in the decoded volumes.

From the different panels, it is possible to see how HetSIREN improves the resolution of different molecule regions, mainly coming from the GR:FKBP51 region, as expected due to its large degree of flexibility. In addition, we show in the last row of the figure an example of the ability of the method to detect small structural details like side chains in contrast with the CryoSPARC maps. It should be noted that the atomic model included in the images has not been refined against the volume-decoded HetSIREN, which translates into small adjustment errors. However, it would

be possible to refine the model against the different HetSIREN conformations to improve their matching.

In addition, we modified Figure 6 to improve the threshold of the maps as suggested:

Figure 6: Resolution analysis of HetSIREN decoded maps compared with the deposited map from (21). Panel a) shows on the left the deposited map (EMD-29069) and the HetSIREN map on the right, both colored by their local resolution estimated with DeepRes (20). The comparison shows an improvement in the local resolution of the map decoded by HetSIREN, mainly occurring in the flexible region composed of GR and FKBP51. Panel b) quantitatively compares the estimated local resolutions based on local resolution histograms. Similarly to panel a), the local resolution of HetSIREN shows a displacement of the voxel resolutions to the high-resolution domain.

It should be noted that, due to the different characteristics of the two volumes, it is hard to find an iso-threshold that displays similar structural features in both maps. We hope the newly selected values simplify their comparison compared to the previous version of the figure.

We include below the new text added to the manuscript for the completeness of the response letter:

In addition, Supplementary Figure 2 presents different zoom regions of the previous two volumes to better compare the differences in the resolution and structural features between the proposed maps. The last row also highlights the capacity of HetSIREN to detect small structural

details like side chains in contrast to the published map. It should be noted that the atomic model used for the comparison was not refined against the HetSIREN decoded volume, translating into small deviations that could be further refined to improve their matching.

For Figure 7 and 8, is there a good reason to arrange the maps by that order? It would be easier to visualize the conformational changes if the authors sort the maps by their locations in the latent space.

We have modified Figures 7 and 8 (which become Figures 8 and 9 in the new manuscript version) by grouping similar conformational changes so they are easier to follow.

Remarks on code availability

I spent some time on this but failed to find the actual implementation of the algorithm in the GitHub repository. The repository provide a wrapper for the HetSiren protocol, but the actual code of that seems to be somewhere else. It would be good if the authors provide a link to the code.

We have included in the “Code availability” section the link to the package “Flexutils-Toolkit” which contains of HetSIREN neural network as follows:

The HetSIREN algorithm is freely available through Scipion 3.0 (17) under the plugin scipion-em-flexutils (<https://github.com/scipion-em/scipion-emflexutils>) and the package Flexutils-Toolkit (<https://github.com/l2PC/Flexutils-Toolkit>). The protocol corresponding to the algorithm described in this manuscript is flexutils - flexible align – HetSIREN.

Reviewer #2

This paper presents the HetSIREN method for effectively achieving multi-conformational 3D reconstruction in cryo-EM. The authors have conducted extensive experiments on various datasets, including simulated and experimental images, to demonstrate the method's effectiveness. They also highlight that HetSIREN can better disentangle different conformations in the latent space by optimizing pose and CTF, while achieving denoising for higher-resolution reconstructions. However, there are still some concerns that need to be addressed.

There is a need for more detailed comparisons and experimental evidence to support its uniqueness, particularly on the new GR:Hsp90:FKBP51 dataset, where analyses using other methods (e.g., cryoDRGN) are lacking.

Following the reviewer's suggestions, we have trained CryoDRGN with the GR:Hsp90:FKBP51 dataset to compare against HetSIREN results. The comparison of the two methods is summarized below:

In both cases, the volumes decoded by CryoDRGN and HetSIREN were taken along the principal component, capturing the motion presented in the manuscript (the first principal component in both cases). As seen from the volumes, both methods properly identify a similar conformational change, but HetSIREN shows a larger amplitude than CryoDRGN. In addition, both maps and slices show a significant difference in the quality of the volumes decoded by the two networks.

In this case, HetSIREN learns a better representation of the structural features of the protein, mainly in the regions exhibiting larger flexibility, while CryoDRGN produces a noisier representation of these regions. This improvement in the quality of the decoded volumes allows us to interpret the conformational change in HetSIREN more accurately than CryoDRGN.

The authors have designed numerous complex loss functions to constrain the training process, but there is a lack of systematic discussion on whether these losses have a meaningful impact on the results. I suggest conducting ablation studies to demonstrate the effectiveness of these losses on prediction outcomes.

Following the reviewer's suggestion, we have further explored and assessed the effect of the additional cost functions in HetSIREN with an ablation test. The results are summarized in a new supplementary section, "Cost function ablation studies."

The first test proposed relies on the 500 adenylate kinase simulated images, starting from the noise-free dataset and then adding noise to analyze the denoising effect more accurately. The results obtained for this test are summarized in Supplementary Figure 11, which is also shown below:

Supplementary Figure 11: Ablation test to analyze the performance of the denoising cost functions implemented during the training phase of HetSIREN. The test evaluates the denoising capabilities of the network under different

noise conditions: a set of ideal images, images with medium noise ($\sigma = 1$), and high noise ($\sigma = 10$). In all cases, two different networks were trained, whose only difference is the presence of the denoising cost functions in one of them. The 3D volumes shown are decoded with the denoising network in all cases.

Two neural networks were trained for each of the three datasets analyzed with variable noise levels: one network does not include the additional denoising cost functions, while the second considers them. As seen from the Figure, even in high noise conditions, HetSIREN learns noise-free and interpretable volumes thanks to the denoising effect of the cost functions described in the manuscript. Moreover, the detected conformational change is the expected one, even in the case of high noise with a reduced number of images in the dataset.

The second test evaluates the effect of the three cost functions in the focused reconstruction step described in the manuscript. Similar to the case before, two networks were trained: one without considering the additional cost functions and the second with the cost functions activated. The dataset used to train the networks was the EMPIAR 10028 dataset, simplifying comparing the results to those already included in the initial version of the manuscript. The results of this test are summarized in Supplementary Figure 12, which is also presented below:

Supplementary Figure 12: Evaluation of the focused reconstruction-related cost functions implemented in HetSIREN. The test evaluates the effect of adding the cost functions responsible for ensuring that the values in chimera volume follow a similar distribution. When this regularization is not applied, the decoded volume shows a clear artifact arising from a strong difference in the value distribution of the refined region and the rest of the volume. In contrast, the regularized network properly minimizes the previous artifact, yielding a more consistent volume.

As seen from the Figure, the main effect of these cost functions is to force the network to learn how to refine a localized region of a map in a heterogeneous manner while considering the characteristics of the voxel values in the original volume. When the cost functions are not considered, the decoded map presents a strong artifact in the range of values learned by the network, as it is free to place any voxel values it considers appropriate to improve the reconstruction loss. However, when the cost functions are considered, the network effectively learns to modify the region of interest while improving the range of voxel values, leading to a more accurate “chimera” volume with no visible artifacts.

We include below the new text added to the manuscript for the completeness of the response letter:

Cost function ablation studies

As discussed in the Methods manuscript section, HetSIREN implements different cost functions directly affecting the decoded volume representation, trying to guide the network toward learning more accurate and interpretable 3D volumes from the images.

One of the effects of the proposed cost functions is to directly tackle the noise present in the images, allowing the network to focus on the signal instead of learning how to add noise to the decoded volumes. To better assess the effect of this denoising, we performed an ablation test starting from the simulated adenylylase kinase images already described above. The tests first analyze the set of noise-free images, followed by a progressive addition of noise. In all these steps, two HetSIREN networks were trained: one did not include the additional denoising cost functions, unlike the second network, which is allowed to learn how to denoise the decoded volumes. It is important to highlight that the three denoising cost functions proposed in the manuscript (L1 regularization and the two versions of the total variation) are evaluated together as they complement each other to reduce the noise while preserving the relevant details.

The results obtained from this analysis are summarized in Supplementary Figure 11. As explained above, the first step is analyzing the original 500 noise-free images. The projections of the decoded volumes show that both HetSIREN networks could identify the correct structure. However, it is possible to observe a non-uniform background when the denoising cost functions are not included, probably generated as a CTF effect. As expected, the network properly detected the conformational change captured by the images.

The analysis continued with a new set of noisy images simulated to have a medium noise level. When medium noise is added, it is possible to observe a more drastic effect on the two neural networks. The network without denoising adds a considerable amount of noise to the decoded volume in an attempt to match the noise of the projection, unlike the network with denoising that manages to produce a noise-free volume similar to one obtained with the baseline images. Similar to the previous case, the detected conformational change is the expected one. Lastly, a dataset with a high level of noise was analyzed. When no denoising is considered, the decoded volumes lack any meaningful signal and are completely dominated by noise. However, the network with denoising manages not only to detect the right signal but also to produce a noise-free decoded volume with features similar to those of the baseline dataset. This result shows the strong effect of handling the noise directly with the network, allowing it to learn accurate and meaningful structure representations even in highly noisy conditions. Moreover, the network with denoising also manages to properly detect the expected conformational change, showing the ability of the network to perform heterogeneous reconstruction with small and noisy datasets.

The next set of cost functions to be evaluated are those related to the focused reconstruction/refinement introduced in the manuscript. The main purpose of these cost functions is to regularize the neural network so that it learns to refine the map while preserving the original voxel value characteristics in the reference volume given to the network. Similar to the case before, it is required to consider these cost functions simultaneously to properly evaluate their effect, as their combination is needed to properly represent the voxel value distribution in the original volume.

For this test, we trained two neural networks using the EMPIAR 10028 dataset to simplify comparing the results with those presented in the rest of the manuscript. The only difference between the two networks trained is the consideration of the cost functions related to the focused reconstruction process. The results from this test are summarized in Supplementary Figure 12. As can be seen from the figure, when the cost functions are not considered, the network introduces a strong artifact in the decoded volume. This artifact arises from the freedom the network has to place any possible voxel value in a given position in the grid, completely breaking the relation of the decoded values with the original distribution of voxel values in the reference volume. In contrast, the regularized network effectively learns to refine the region of interest while considering that the range of values of the decoded region should be as similar as

possible to the reference volume. Thus, the regularized network does not present the artifact previously described, improving the representation and interpretability of the decoded volume.

The authors emphasize the significant denoising effects of their method. However, the paper lacks a convincing discussion of the denoising results. It is known that density maps generally require sharpening after 3D reconstruction, or post-processing using current deep learning methods like DeepEMhancer and EMReady. These deep learning post-processing methods largely achieve denoising effects. Does the authors' claim imply that HetSIREN has integrated a full pipeline of 3D reconstruction with deep learning-based post-processing? I would be very interested to see a comparison between the reconstructions using HetSIREN and those using cryoSPARC with other methods like DeepEMhancer and/or EMReady.

In HetSIREN, we have integrated an internal sharpening process that relies on a combined effect of the cost functions and the volume generation process, which allows us to directly generate a deconvolved representation of the volume by analyzing the Gaussian coefficients representing the volume decoded by the network. However, this internal sharpening does not prevent further postprocessing of a HetSIREN volume with popular sharpening software like DeepEMHancer or EMReady. In the specific case of DeepEMHancer, even if it has not been trained with the maps produced by HetSIREN, it has always produced satisfactory results in our hands when applied to different datasets that we have analyzed with HetSIREN.

To better illustrate the previous concepts and follow the reviewer's suggestions, we have included a new supplementary section, "Cost function ablation studies," with a short discussion on the sharpening effect and application in HetSIREN compared to CryoSPARC. The results are summarized in Supplementary Figure 12, which is also presented below:

Supplementary Figure 12: Comparison of HetSIREN and CryoSPARC reconstruction for the EMPIAR 10028 dataset. The comparison shows first the original volumes obtained by both approaches on the left. In the case of HetSIREN, the decoded volume includes the internal sharpening applied during the decoding step, as described in the manuscript. In addition, the figure shows the previous two volumes further post-processed by DeepEMHancer (45) to further enhance their structural features. This comparison reveals that the internal sharpening implemented in HetSIREN does not prevent further modification of the decoded volume, yielding a new representation with significantly enhanced structural features compared to both the CryoSPARC maps and its sharpened representation.

As seen from the Figure, HetSIREN's internal sharpening leads to a higher quality volume than the CryoSPARC refined volume. Let's compare the volume obtained after applying DeepEMhancer to the CryoSPARC map against the volume decoded directly by HetSIREN. It is possible to appreciate that they have similar features and quality. It is essential to highlight that this similarity appears even if we compare it against the volume directly decoded by HetSIREN, which only has the internal sharpening implemented in the network.

As commented before, sharpening software like DeepEMhancer can be used to post-process the HetSIREN volume further. Thus, we executed DeepEMhancer with the HetSIREN decoded map to produce a sharpened volume. Comparing the HetSIREN and CryoSPARC sharpened maps shows a clear difference in the enhancement of the structural features, which are more prominent in the case of the post-processed HetSIREN volume.

We include below the new text added to the manuscript for the completeness of the response letter:

Apart from the denoising and focused reconstruction-related losses previously evaluated, HetSIREN includes an internal sharpening arising as a post-processing effect from the way the decoded volumes are constructed, which is added to the enhancement effects of the additional cost functions. However, this internal sharpening does not prevent further post-processing of the decoded volume to further enhance the structural features in the volume, which is an essential step to properly understand and interpret a given biomolecular structure.

To better reflect the previous idea, we performed a comparison of HetSIREN with the internal sharpening and a further post-processed version of the decoded volume with DeepEMhancer (45). To that end, we compared one of the HetSIREN volumes decoded for the EMPIAR-10028 dataset previously discussed in the manuscript. The comparison is presented in Supplementary Figure 12. We present as a baseline of the comparison the CryoSPARC volume reconstructed from this dataset. The comparison shows how the internal sharpening of HetSIREN significantly improved the structural features in the volume, which are similar to the ones present in the CryoSPARC volume post-processed with DeepEMhancer. In addition, the sharpening post-processing of the HetSIREN volume enhances even further the structural features compared to its non-sharpened version and the sharpened CryoSPARC volume.

RESPONSE LETTER

In the following document, we include our detailed answers to the reviewer's comments, including a description of the modifications in the manuscript to simplify their tracking in the new version. The letter is formatted so the reviewer's comments are highlighted in blue, followed by the answers in dark green. For the completeness of the response letter, we also include the original text included in the reviewed version of the manuscript in purple at the end of every answer.

Reviewer #1

Remarks to the Author

The authors have carefully addressed all my comments in the revised manuscript, and the additional experiments nicely demonstrated the performance of the individual modules of HetSIREN. I am overall happy with the revised version and recommend the publication of the paper.

In supplementary figure 9, the text in the figure says CTF decoupled while the caption talks about pose decoupling.

We have modified Supplementary Figure 9 to correct the typo. The new Figure is shown below:

Consider using a brighter colour scheme for Figure 6a. Currently the high-resolution part of the protein is so dark the features are hardly distinguishable.

We have increased the colour scheme brightness as suggested by the reviewer in Figure 6a to improve the visibility of the map features. The new Figure is included below:

In supplementary figure 2, if the authors claim the extra density to be a sidechain, it is necessary to show the sidechain atoms of the model and have it roughly fit into the density.

We have modified Supplementary Figure 2 to include the side chains in the displayed model as suggested. The new Figure is shown below:

HetSIREN

CryoSPARC

Remarks on code availability

The code seems reasonably well-arranged. I still have not found a tutorial on how to use the method. A link to such a tutorial in the manuscript would be helpful.

We have added to the Code Availability section a link to the tutorial of HetSIREN. The new text reads:

Tutorials on how to setup and use HetSIREN are provided in the following webpage:
https://scipion-em.github.io/docs/release-3.0.0/docs/user/tutorials/flexibilityHub/main_page.html#tutorials

Reviewer #2

The authors have added detailed comparative results related to cryoDRGN and DeepEMHancer, which addressed most of my comments. It would also be valuable to give a comparison with EMReady or a discussion on the topic as it has been widely used in the cryo-EM field. Other than that, I have no more comments.

Following the reviewer's suggestions, we have trained CryoDRGN with the GR:Hsp90:FKBP51 dataset to compare against HetSIREN results. The comparison of the two methods is summarized below:

Supplementary Figure 13: Comparison of HetSIREN and CryoSPARC reconstruction for the EMPIAR 10028 [2] dataset. The comparison shows first the original volumes obtained by both approaches. In the case of HetSIREN, the decoded volume includes the internal sharpening applied during the decoding step, as described in the manuscript. In addition, the figure shows the previous two volumes further post-processed by DeepEmhancer [17] and EMReady [18] to further enhance their structural features. This comparison reveals that the internal sharpening implemented in HetSIREN does not prevent further modification of the decoded volume, yielding a new representation with significantly enhanced structural features compared to both the CryoSPARC maps and its sharpened representation.